# BEAM TREE RECURSIVE CELLS

## ABSTRACT

Recursive Neural Networks (RvNNs) generalize Recurrent Neural Networks (RNNs) by allowing sequential composition in a more flexible order, typically, based on some tree structure. While initially user-annotated tree structures were used, in due time, several approaches were proposed to automatically induce tree-structures from raw text to guide the recursive compositions in RvNNs. In this paper, we present an approach called Beam Tree Recursive Cell (or BT-Cell) based on a simple yet overlooked backpropagation-friendly framework. BT-Cell adapts beam search easy-first parsing for simulating RvNNs with automatic structure-induction. Our results show that BT-Cell achieves near-perfect performance on several aspects of challenging structure-sensitive synthetic tasks like ListOps and also comparable performance in realistic data to other RvNN-based models. We further introduce and analyze several extensions of BT-Cell based on relaxations of the hard top-k operators in beam search. We evaluate the models in different out of distribution splits in both synthetic and realistic data. Additionally, we identify a previously unknown failure case for neural models in generalization to unseen number of arguments in ListOps. Code is in the supplementary.

## 1 INTRODUCTION

In the space of sequence encoders, Recursive Neural Networks (RvNNs) can be said to lie somewhere in-between Recurrent Neural Networks (RNNs) and Transformers in terms of flexibility. While vanilla Transformers show phenomenal performance and efficient scalability on a variety of tasks, it can often struggle in length generalization and systematicity in syntax-sensitive tasks (Tran et al., 2018; Shen et al., 2019a; Lakretz et al., 2021; Csordás et al., 2022). RvNN-based models, on the other hand, can often excel on some of the latter kind of tasks (Shen et al., 2019a; Chowdhury & Caragea, 2021; Liu et al., 2021; Bogin et al., 2021) making them worthy of further study although they may suffer from limited scalability in their current formulations.

Given an input text, RvNNs (Pollack, 1990; Socher et al., 2010) are designed to build up the representation of the whole text by recursively building up the representations of its constituents starting from the most elementary representations (tokens) in a bottom-up fashion. As such, RvNNs can model the hierarchical part-whole structures underlying texts. However, originally RvNNs required access to pre-defined hierarchical constituency-tree structures. Several works (Socher et al., 2011; Havrylov et al., 2019; Choi et al., 2018; Maillard et al., 2019; Chowdhury & Caragea, 2021) introduced latent-tree RvNNs that sought to move beyond this limitation by making RvNNs able to learn to automatically determine the structure of composition from any arbitrary downstream task objective, given just the raw input text.

Among these approaches, Gumbel-Tree models (Choi et al., 2018) are particularly attractive for its simplicity. It often serves as a standard baseline for latent-tree models. However, Gumbel-Tree models not only suffer from biased gradients (due to use of Straight-Through Estimation (STE)), but they also perform poorly on synthetic tasks like ListOps (Nangia & Bowman, 2018) that were specifically designed to diagnose the capacity of neural models for automatically inducing underlying hierarchical structures. In this paper, we tackle these issues by introducing the Beam Tree Cell (BT-Cell) framework that applies beam-search as a simple modification over Gumbel-Tree models. Instead of greedily selecting the highest scored sub-tree representations like Gumbel-Tree models, BT-Cell chooses and maintains top-$k$ highest scored sub-tree representations. We show that this simple modification increases the performance of Gumbel-Tree models in challenging structure sensitive tasks by several folds. For example, in ListOps, when testing for samples of length 900-1000, a BT-Cell

based model increases the performance of a comparable Gumbel-Tree model from 37.9% to 86.7% (see: Table 1). We further explore several variants of BT-Cell. Particularly, we explore ways to replace the non-differentiable top-k operators involved in beam search with different alternatives such as top-k gumbel softmax with STE and a novel strategy of maintaining a convex combination of bottom scoring paths. Our best extension achieves a new state-of-the-art in length generalization and depth-generalization in structure-sensitive synthetic tasks like ListOps and performs comparably in realistic data against other latent-tree models.

A few recently proposed latent-tree models simulating RvNNs like LSTM-RL (Havrylov et al., 2019), Ordered Memory (OM) (Shen et al., 2019a) or CRvNN (Chowdhury & Caragea, 2021) are also strong contenders to BT-Cell and its extensions on synthetic data. However, unlike BT-Cell, LSTM-RL relies on expensive reinforcement learning and several sophisticated techniques to stabilize training. Moreover, compared to OM and CRvNN, one distinct advantage of BT-Cell is that it not just provides the final sequence encoding (representing the whole input text) but also the intermediate constituent representations at different levels of hierarchy (representations of all nodes of the underlying induced trees). Such tree-structured node representations can be useful as inputs to further downstream modules like a Transformer (Vaswani et al., 2017) or GNN (Scarselli et al., 2009) in a full end-to-end setting [1]. While CYK-based RvNNs (Maillard et al., 2019) are also promising and similarly can provide multiple span representations they tend to be much more expensive than BT-Cell. All these architectural trade-offs among different latent-tree models are discussed in more details in Appendix E.6.

Besides proposal and evaluation of BT-Cell variants, our paper also serves as a survey of how well prior proposed latent-tree RvNNs work in structure-sensitive synthetic tasks and out-of-distribution-splits in natural language tasks, particularly when combined with more powerful recursive cells. Additionally, as a further contribution, we identify a previously unknown failure case for even the best performing neural models when it comes to argument generalization in ListOps (Nangia & Bowman, 2018) - opening up a new challenge for future research.

## 2 PRELIMINARIES

**Problem Formulation:** Similar to Choi et al. (2018), throughout this paper, we explore the use of RvNNs as a sentence encoder. Formally, given a sequence of token embeddings $\mathcal{X} = (e_1, e_2, \ldots, e_n)$ (where $\mathcal{X} \in \mathbb{R}^{n \times d_e}, e_i \in \mathbb{R}^{d_e}$, and $d_e$ is the embedding size), the task of a sentence encoding function $\mathcal{E} : \mathbb{R}^{n \times d_e} \to \mathbb{R}^{d_h}$ is to encode the whole sequence of vectors into a single vector $o = \mathcal{E}(\mathcal{X})$ (where $o \in \mathbb{R}^{d_h}$ and $d_h$ is the size of the encoded vector). We can use a sentence encoder for sentence-pair comparison tasks like logical inference or for text classification.

### 2.1 RECCURENT NEURAL NETWORKS AND RECURSIVE NEURAL NETWORKS

A core component of both RNNs and RvNNs is a recursive cell. In our contexts, the cell function takes as arguments two vectors ($a_1 \in \mathbb{R}^{d_{a_1}}$ and $a_2 \in \mathbb{R}^{d_{a_2}}$) and returns a single vector $v = cell(a_1, a_2)$ (where $v \in \mathbb{R}^{d_v}$). $cell : \mathbb{R}^{d_{a_1}} \times \mathbb{R}^{d_{a_2}} \to \mathbb{R}^{d_v}$. In our settings, we generally set $d_{a_1} = d_{a_2} = d_v = d_h$. Given a sequence $\mathcal{X}$, both RNNs and RvNNs sequentially process it through recursive application of the cell function. For a concrete example, consider a sequence of token embeddings such as $(2 + 4 \times 4 + 3)$ (Assume the symbols 2, 4, + etc. represent transformations of corresponding embedding vectors $\in d_h$). Given any such sequence, RNNs can only follow a fixed left-to-right order of composition. For the particular aforementioned sequence, an RNN-like application of the cell function can be expressed as:

$$o = cell(cell(cell(cell(cell(cell(cell(h0, 2), +), 4), \times), 4), +), 3) \quad (1)$$

Here $h0$ is some input-independent initial state ("initial hidden state") set in the model. In contrast to RNNs, RvNNs can compose the sequence in more flexible orders. For example, one way (among

---

[1] There are several works that have used intermediate span representations for better compositional generalization in generalization tasks (Liu et al., 2020; Herzig & Berant, 2021; Bogin et al., 2021; Liu et al., 2021; Mao et al., 2021). We keep it as a future task to explore whether the span representations returned by BT-Cell can be used in relevant ways.

many) that RvNNs could apply the cell function is as follows:

$$o = cell(cell(cell(cell(2, +), cell(cell(4, \times), 4)), +), 3) \tag{2}$$

Thus, RNNs can be considered as, roughly, a special case of RvNNs where a strict left-to-right order of composition is enforced. As we can see, by these strategies of recursively reducing two vectors into a single vector, both RNNs and RvNNs can implement the sentence encoding function in the form of $\mathcal{E}$. Moreover, the form of application of cell function exhibited by RNNs and RvNNs can also be said to reflect a tree-structure. For any application of the cell function in the form $v = cell(a_1, a_2)$, $v$ can be treated as the representation of the immediate parent node of child nodes $a_1$ and $a_2$ in a underlying tree.

In Eqn. 2, we find that RvNNs can align the order of composition to PEMDAS whereas RNNs cannot. Nevertheless, RNNs can still learn to simulate RvNNs by modeling tree-structures implicitly in their hidden state dimensions (Bowman et al., 2015b). For example, RNNs can learn to hold off the information related to "2+" until "$4 \times 4$" is processed. Their abilities to handle tree-structures is analogous to how we can use pushdown automation in a recurrent manner through an infinite stack to detect tree-structured grammar. Still, RNNs can struggle to effectively learn to appropriately organize information in practice for large sequences. Special inductive biases can be incorporated to enhance their abilities to handle their internal memory structures (Shen et al., 2019b;a). However, even then, memories remain bounded in practice and there is a limit to what depth of nested structures they can model.

More direct approaches to RvNNs, in contrast, can alleviate the above problems and mitigate the need of sophisticated memory operations to arrange information corresponding to a tree-structure because they can directly compose according to the underlying structure (Eqn. 2). However, in the case of RvNNs, we have the problem of first determining the underlying structure to even start composition. One approach to handle the issue can be to train a separate parser to induce a tree structure from sequences using gold tree parses. Then we can use the trained parser in RvNNs. However, this is not ideal. Not all tasks or languages would come with gold trees for training a parser and a parser trained in one domain may not translate well to another. A potentially better approach is to jointly learn both the cell function and structure induction from a downstream objective. We focus on this latter approach. Below we discuss one framework for this approach.

## 2.2 EASY-FIRST PARSING AND GUMBEL TREE MODELS

Here, we describe an adaptation (Choi et al., 2018) of easy-first parsing (Goldberg & Elhadad, 2010) for RvNN-based sentence-encoding. The algorithm relies on a scorer function $score : \mathbb{R}^{d_h} \to \mathbb{R}^1$ that scores parsing decisions. Particularly, if we have $v = cell(a_1, a_2)$, then $score(v)$ represents the plausibility of $a_1$ and $a_2$ belonging to the same immediate parent constituent. In practice, similar to Choi et al. (2018), we keep the scorer as a simple linear transformation: $score(v) = W_v v$ (where $W_v \in \mathbb{R}^{1 \times d_h}$ and $v \in \mathbb{R}^{d_h}$).

**Recursive Loop:** In this algorithm, at every iteration in a recursive loop, given a sequence of hidden states $(h_1, h_2, \ldots, h_n)$ we consider all possible immediate candidate parent compositions taking the current states as children: $(cell(h_1, h_2), cell(h_2, h_3), \ldots, cell(h_{n-1}, h_n))^2$. We then score each of the candidates with the score function and greedily select the highest scoring candidate (i.e. we commit to the "easiest" decision first). For the sake of illustration, assume $score(cell(h_i, h_{i+1})) \geq score(cell(h_j, h_{j+1})) \ \forall j \in \{1, 2, \ldots, n\}$. Thus, following the algorithm the parent candidate $cell(h_i, h_{i+1})$ can be chosen. The parent representation $cell(h_i, h_{i+1})$ would then replace its immediate children $h_i$ and $h_{i+1}$. Thus, the resulting sequence will become: $(h_1, \ldots, h_{i-1}, cell(h_i, h_{i+1}), h_{i+2}, \ldots, h_n)$. Like this, the sequence will be iteratively reduced to a single element representing the final sentence encoding. The full algorithm is presented in the Appendix (see Algorithm 1).

One issue here is how to choose the highest scoring candidate. One way to do it is to simply use an argmax operator but it will not be differentiable. Gumbel-Tree-Cell models (Choi et al., 2018) address this difficulty by using Straight Through Estimation (STE) (Bengio et al., 2013) with Gumbel Softmax (Jang et al., 2017; Maddison et al., 2017) instead of argmax. However, STE is

---

[2]We focus only on the class of binary projective tree structures. Thus all the candidates are compositions of two contiguous elements.

known to cause high bias in gradient estimation. Moreover, as it was previously discovered (Nangia & Bowman, 2018), and as we independently verify, STE Gumbel-based strategies perform poorly when tested in structure-sensitive tasks. Instead, to overcome these issues, we present an alternative of extending argmax with a top-k operator under a beam search strategy.

## 3 BEAM TREE CELL

**Motivation:** Gumbel-Tree models, as described, are relatively fast and simple but they are fundamentally based on a greedy algorithm for a task where the greedy solution is not guaranteed to be optimal. On the other hand, adaptation of dynamic programming-based CYK-models (Maillard et al., 2019) leads to high computational complexity (discussed more in Appendix E.6). A "middle way" between the two extremes is then to simply extend Gumbel-Tree models with beam-search to make it less greedy while still being less costly than CYK-parsers (See Appendix E.6)). Moreover, Using beam-search also provides additional opportunity to recover from local errors whereas a greedy single-path approach (like Gumbel Tree models) will be stuck with any errors made. All these factors motivate the framework of Beam Tree Cells (BT-Cell).

**Implementation:** The beam search extension to Gumbel-Tree models is straight-forward and similar to standard beam search. The method is described more precisely in Appendix A.1 and Algorithm 2. In summary, in BT-Cell, given a beam size $k$, we maintain a maximum of $k$ hypotheses (or beams) at each recursion. In any given iteration, each beam constitutes a sequence of hidden states representing a particular path of composition and an associated score for that beam based on the addition of log-softmaxed outputs of the $score$ function (as defined in 2.2) over each chosen compositions for that sequence. At the end of the recursion, we will have $k$ sentence encodings $((o_1, o_2, \ldots, o_k)$ where $o_i \in \mathbb{R}^{d_h})$ and their corresponding scores $((s_1, s_2, \ldots, s_k)$ where $s_i \in \mathbb{R}^1)$. The final sequence encoding can be then represented as: $\sum_{i=1}^{k} \left( \frac{exp(s_i) \cdot o_i}{\sum_{i=1}^{k} exp(s_i)} \right)$. This aims at computing the expectation over the $k$ sequence encodings.

### 3.1 TOP K VARIANTS

As in standard beam search, BT-Cell requires two top-k operators. The first top-k replaces the straight-through gumbel softmax (simulating top-1) in Gumbel-Tree models. However, selecting and maintaining $k$ possible choices for every beams in every iteration leads to an exponential increase in the number of total beams. Thus, a second top-k operator is used for pruning the beams to maintain only a maximum of $k$ beams at the end of each iteration. Now, different variants of BT-Cell can be established depending on what kind of top-k operator we use.

**Plain Top-k:** The simplest variant is to simply use the vanilla top-k operator. However, the vanilla top-k operator is discrete and non-differentiable preventing gradient propagation to non-selected paths. Despite that this can still work for the following reasons: (1) Gradients can still pass through the final top $k$ beams and scores. The scorer function can thus learn to increase the scores of better beams and lower the scores of the worse ones among the final $k$ beams; (2) A rich enough cell function can be robust to local errors in the structure and learn to adjust for it by organizing information better in its hidden states. We believe that as a combination of these two factors, plain BT-Cell even with non-differentiable top-k operators can learn to perform well for structure-sensitive tasks (as we will empirically observe).

**ST-Gumbel Top-k:** While non-differentiable top-k operators can work, they still can be a bottleneck because gradient signals will be received only for $k$ beams in a space of exponential possibilities. To address this, we replace the plain top-k operator with a STE (Bengio et al., 2013) through Gumbel top-k (Kool et al., 2019). We refer to this operator as ST-Gumbel Top-k. This serves as a natural generalization of the original straight-through gumbel softmax from Choi et al. (2018) (where we had to select only one item) to the top-k context[3]. However, ST-Gumbel Top-K can exacerbate the original issues of biased estimation related to straight-through methods (we discuss more in Appendix D).

---

[3]In practice we find it beneficial to only replace the first top-k operator with ST-gumbel Top-k. Replacing the second top-k operator with ST-Gumbel top-k tends to lead to instability or worse performance.

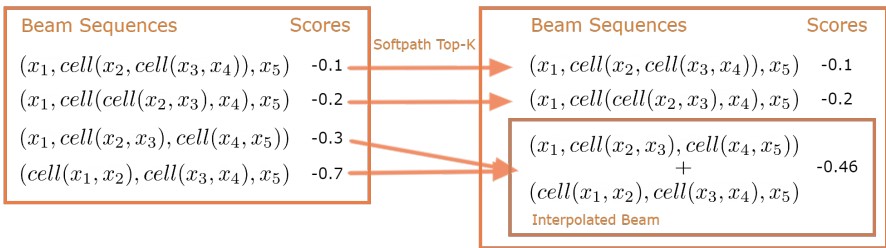

Figure 1: Visualization of Top-k Softpath selection from $m = 4$ beams to top $k = 3$ beams.

**Softpath Top-k:** Here, we focus on the second top-k operator that is involved in truncating beams. As a concrete case, assume we have $m$ beams (sequences and their corresponding scores). The target for a top-k operator is to keep only the top scoring $k$ beams (where $k \leq m$).

Ideally we want to keep the beam representations "sharp" and avoid washed out representations owing to interpolation (weighted vector averaging) of distinct paths (Drozdov et al., 2020). This can be achieved by either plain top-k or ST-Gumbel top-k. However, the former prevents propagation of gradient signals through the bottom $m - k$ beams, and the latter can exacerbate bias (see Appendix D) as discussed before. Another line of approach is to create a soft permutation matrix $P \in \mathbb{R}^{m \times m}$ through a differentiable sorting algorithm such that $P_{ij}$ represents the probability of the $i^{th}$ beam being the $j^{th}$ highest scoring beam. $P$ can then be used to sofity select the top $k$ beams. However, running differentiable sorting in a recursive loop can significantly increase computation overheads and also create more "washed out" representations leading to higher error accumulation (we discuss more in Appendix E.1). We tackle all these challenges by instead proposing a simple hybrid strategy to approach top-k selection. We provide a formal description of our proposed strategy below and a visualization of the process in Figure 1.

Assume we have $m$ beams consisting of $m$ sequences: $H = (\mathcal{H}_1, \ldots, \mathcal{H}_m)$ ($\mathcal{H}_i \in \mathbb{R}^{n \times d_h}$ and $n$ being the sequence length) and $m$ corresponding scores: $S = (s_1, \ldots, s_m)$. First, we simply use the plain top-k operator to discretely select the top $k - 1$ beams (instead of $k$). This allows us to keep the most promising beams "sharp":

$$idx = topk(S, K = k - 1). \quad Top = \{(\mathcal{H}_i, s_i) \mid i \in idx\} \tag{3}$$

Second, for the $k^{th}$ beam we instead perform a softmax-based marginalization of the bottom $m - (k - 1)$ beams. This allows us to still propagate gradients through the bottom scoring beams (unlike in the pure plain top-k operator):

$$Bottom = \{(\mathcal{H}_i, s_i) \mid (i \notin idx) \wedge (i \in \{1, 2, \ldots, m\})\} \tag{4}$$

$$softpath = \left( \sum_{(\mathcal{H},s) \in Bottom} \left( \frac{exp(s) \cdot \mathcal{H}}{\sum_{(\mathcal{H},s) \in Bottom} exp(s)} \right), \sum_{(\mathcal{H},s) \in Bottom} \left( \frac{exp(s) \cdot s}{\sum_{(\mathcal{H},s) \in Bottom} exp(s)} \right) \right) \tag{5}$$

Finally we add the softpath to the top $k - 1$ discretely selected beams to get the final set of $k$ beams: $Top \cup \{softpath\}$. Thus, we get to achieve a "middle way" between plain top-k and differentiable sorting: partially getting the benefit of sharp representations of the former through discrete top $k - 1$ selection, and partially getting the benefit of gradient propagation of the latter through soft-selection of the $k^{th}$ beam. In practice, we find it beneficial to switch to plain top-k during inference.

**Gumbelpath Top-k:** Here, we replace the softmax in the softpath extension (eqn 5) with a straight-through gumbel softmax making the induced structure discrete during training as well.

## 4 EXPERIMENTS AND RESULTS

Hyperparameters and other architectural details are in the Appendix H. Next, we discuss the main models that we compare.

**1. RecurrentGRC:** RecurrentGRC is an RNN implemented with the Gated Recursive Cell (GRC) (Shen et al., 2019a) as the cell function (see Appendix B for description of GRC).

**2. RandomTreeGRC:** RandomTreeGRC is an RvNN with GRC that randomly chooses parent compositions at each iteration (based on uniform probabilities).

**3. BalancedTreeGRC:** BalancedTreeGRC is an RvNN with GRC that enforces a binary balanced tree structure. Some prior experiments (Shi et al., 2018) have shown strong results just by using balanced trees like this.

**4. GoldTreeGRC:** GoldTreeGRC is a GRC-based RvNN with gold tree structures.

**5. GumbelTreeLSTM:** This is the straight-through Gumbel-tree implementation by Choi et al. (2018). It uses the LSTM cell (Hochreiter & Schmidhuber, 1997; Tai et al., 2015).

**6. GumbelTreeGRC:** This is same as GumbelTreeLSTM but with GRC instead of LSTM.

**7. BSRP-GRC:** Maillard & Clark (2018) proposed a similar approach to ours where they apply beam search into a shift-reduce parsing framework instead of easy-first-parsing. We adapt that framework with the GRC cell and call it as BSRP-GRC. More details are provided in Appendix C.

**8. CYK-GRC:** This is the CYK-based model proposed by Maillard et al. (2019) but with GRC.

**9. Ordered Memory:** This is a form of memory-augmented RNN simulating certain classes of RvNN functions as proposed by (Shen et al., 2019a). Ordered Memory also uses GRC.

**10. CRvNN:** CRvNN is a variant of RvNN with a continuoux relaxation over its structural operations as proposed by Chowdhury & Caragea (2021). CRvNN also uses GRC. In our implementation, we ignore some extraneous elements from CRvNN such as transition features and halt penalty which were deemed to have little effect during ablation.

**11. BT-LSTM:** Base BT-Cell model with LSTM cell and plain top-k operator.

**12. BT-GRC:** Base BT-Cell model with GRC and plain top-k operator.

**13. Gumbel BT-GRC:** BT-Cell model with GRC and ST-Gumbel top-k operator.

**14. BT-GRC + Softpath:** BT-Cell model with GRC and Softpath top-k operator.

**15. BT-GRC + Gumbelpath:** BT-Cell model with GRC and Gumbelpath top-k operator.

For experiments with BT-Cell models, we consider beam size 5 as a practical choice which is neither too big nor too small. However, we also explore beam size 2 for the most promising variants of BT-Cell to see how far we can get with the minimally costly version of beam search.

## 4.1 LISTOPS LENGTH GENERALIZATION RESULTS

**Dataset Settings:** ListOps Nangia & Bowman (2018) is a challenging synthetic task that requires solving of nested mathematical operations over lists of arguments. We present our results on ListOps in Table 1. To test for length-generalization performance, we train the models only on sequences with $\leq 100$ lengths (we filter the rest) and test on splits of much larger lengths (eg. $200 - 300$ or $900 - 1000$) taken from Havrylov et al. (2019). "Near-IID" is the original test set of ListOps (it is "near" IID and not fully IID because a percentage of the split has $> 100$ length sequences whereas such lengths are absent in the training split).

**Results: 1. Heuristic Tree Models:** As discussed before in §2.1, RNNs has to model tree structures implicitly in their bounded hidden states and thus can struggle generalizing to unseen structural depths. This is reflected in the sharp degradation in its length generalization performance. Unsurprisingly, other heuristic-tree-based models (BalancedTreeGRC or RandomTreeGRC) do not perform well either in this structure-sensitive task. **2. Gumbel Tree Models:** Consistent with prior work Nangia & Bowman (2018), Gumbel-Tree models fail to perform well in this task; likely, due to its biased gradient estimation. **3. CYK-GRC:** CYK-GRC shows some promise to length generalization but it was too slow to run in higher lengths (see discussion in Appendix E.6). **4. Ordered Memory (OM):** Here, we find OM struggles to generalize to higher unseen lengths. OM's reliance of soft sequential updates in a nested loop can lead to higher error accumulation over larger unseen lengths or depth. **5. CRvNN:** Consistent with Chowdhury & Caragea (2021), CRvNN performs

| Model | near-IID | Length Gen. | | | Argument Gen. | | LRA |
|---|---|---|---|---|---|---|---|
| (Lengths) | $\leq 1000$ | 200-300 | 500-600 | 900-1000 | 100-1000 | 100-1000 | 2000 |
| (Arguments) | $\leq 5$ | $\leq 5$ | $\leq 5$ | $\leq 5$ | 10 | 15 | 10 |
| *With gold trees* | | | | | | | |
| GoldTreeGRC | 99.95 | 99.88 | 99.85 | 100 | 80.5 | 79 | 78.1 |
| *Baselines without gold trees* | | | | | | | |
| LSTM-RL* | $99.2_5$ | — | — | — | — | — | — |
| RecurrentGRC | 84.05 | 33.85 | 20.2 | 15.1 | 37.35 | 30.10 | 20.7 |
| BalancedTreeGRC | 59.4 | 44.85 | 43.35 | 35.70 | 45.88 | 45.25 | 41.95 |
| RandomTreeGRC | 70.56 | 48.70 | 45.35 | 37.53 | 54.8 | 55.6 | 49.8 |
| GumbelTreeLSTM | 63.08 | 45.8 | 42.75 | 36.8 | 48.95 | 49.2 | 45.35 |
| GumbelTreeGRC | 74.89 | 47.6 | 43.85 | 37.9 | 51.35 | 50.5 | 46.1 |
| CYK-GRC | 97.87 | 93.75 | — | — | 60.75 | 42.45 | — |
| BSRP-GRC | 70.34 | 42.4 | 33.15 | 26.3 | 40.15 | 35.75 | 29.65 |
| Ordered Memory | 99.88 | **99.55** | 92.7 | 76.9 | **84.15** | **75.05** | **80.1** |
| CRvNN† | $99.6_3$ | $98.51_{11}$ | $97.95_{11}$ | $96.78_{19}$ | — | — | — |
| CRvNN | 98.86 | 95.89 | 93.15 | 89.4 | 57.8 | 24.35 | 45.1 |
| *Beam Tree Models with beam size 5 (also without gold trees)* | | | | | | | |
| BT-LSTM | 94.11 | 85.1 | 83.5 | 78.8 | 67.9 | 44.25 | 57.85 |
| BT-GRC | 99.39 | 96.15 | 92.55 | 86.7 | 77.1 | 63.7 | 67.3 |
| Gumbel-BT-GRC | 96.15 | 75.35 | 64.25 | 55.1 | 58.65 | 50.4 | 50.2 |
| BT-GRC + Softpath | **99.92** | 99.5 | 99 | 97.2 | 76.05 | 67.9 | 71.8 |
| BT-GRC + Gumbelpath | 99.84 | 99.45 | **99.2** | **99.4** | 79.25 | 63 | 72.85 |
| *Beam Tree Models with beam size 2 (also without gold trees)* | | | | | | | |
| BT-GRC | 94.18 | 68.2 | 56.85 | 50.2 | 64.45 | 56.95 | 55.85 |
| BT-GRC + Softpath | 99.69 | 97.55 | 95.40 | 91 | 75.75 | 62 | 66.1 |
| BT-GRC + Gumbelpath | 88.56 | 55.4 | 51.3 | 46.1 | 54.65 | 52.95 | 49.55 |

Table 1: Accuracy on ListOps. * indicates results from Havrylov et al. (2019). † indicates results from Chowdhury & Caragea (2021). LSTM-RL is the model proposed in Havrylov et al. (2019). For our models we report the median of 3 runs. Our models were trained on lengths $\leq 100$, depth $\leq 20$, and arguments $\leq 5$. We bold the best results and underline the second-best among models that do not use gold trees. Subscript represents standard deviation. As an example, $90_1 = 90 \pm 0.1$ .

relatively decently at higher lengths. **6. BSRPC-GRC:** Surprisingly, despite using a similar framework to BT-Cell, BSRPC-GRC performs quite poorly. We suspect this is because of the limited gradient signals from its top-k operators coupled with the doubling of recurrent steps (that can cause gradient issues) due to taking a shift-reduce strategy. Moreover, BSRPC-GRC. unlike BT-Cell, also lacks the global competition among all parent compositions when making shift/reduce choices. **7. BT-GRC and BT-LSTM:** Here, we find a massive boost over Gumbel-tree baselines even when using the base models: BT-GRC or BT-LSTM (beam size 5). In the 900-1000 length generalization split, BT-GRC increases the performance of GumbelTreeGRC from $37.9\%$ to $86.7\%$ - all just by adding beam search with plain top-k. But as expected, the performance degrades severely when the beam size is reduced to 2. Note that the recurrent depth for BT-Cell as it pertains to backpropagation is just the tree depth (not the doubled sequence length as in BSRP-GRC). This may further explain its superiority to BSRP-GRC. **8. Gumbel BT-GRC:** This model does not perform as good. Part of its issue could be related to, again, the biased estimation due to STE. We also discuss more specific issues that applies to this model but not Gumbelpath in Appendix D. **9. Softpath and Gumbelpath:** Softpath was specifically designed to counteract the bottleneck of gradient propagation being limited through only $k$ beams in the base BT-Cell (with plain top-k). The bottleneck is even worse when $k$ (beam size) is 2 - empirically this is reflected in the poor performance of base BT-GRC with beam size 2. However, Softpath does what it's designed to: it counteracts the bottleneck. Empirically, this is reflected in its much higher performance than base BT-GRC under beam size 2. Where base BT-GRC gets only $50.2\%$ in the $900 - 1000$ split, BT-GRC+Softpath gets $91\%$. BT-GRC+Softpath has near perfect length generalization with beam size 5. The performance of Gumbelpath is more

| Model | SST2 | SST5 | | IMDB | | |
|-------|------|------|------|------|------|------|
| | IID | IID | LG | IID | Con. | Count. |
| RecurrentGRC | $89.44_{0.5}$ | $52.19_{1.5}$ | $47.45_{3.9}$ | $90.94_{1.2}$ | $74.86_{28}$ | $82.72_{19}$ |
| BalancedTreeGRC | $87.83_{3.7}$ | $\underline{52.35_{6.2}}$ | $46.37_{3.9}$ | $90.71_{0.9}$ | $74.93_{22}$ | $83.61_{15}$ |
| RandomTreeGRC | $89.33_{1}$ | $51.78_{1.2}$ | $49.83_{1.2}$ | $\underline{91.68_{1.1}}$ | $74.93_{14}$ | $82.38_{9.3}$ |
| GumbelTreeLSTM | $88.65_{2.1}$ | $\mathbf{52.44_8}$ | $48.43_{6.3}$ | $88.39_{6.9}$ | $72.27_{15}$ | $80.67_{13}$ |
| GumbelTreeGRC | $89.22_{5.6}$ | $51.67_{8.8}$ | $\mathbf{50.3_{6.3}}$ | $85.11_{11}$ | $70.63_{21}$ | $81.97_5$ |
| CYK-GRC | $\mathbf{89.66_{5.8}}$ | $51.99_{13}$ | $49.14_{3.3}$ | OOM | OOM | OOM |
| Ordered Memory | $\underline{89.46_{1.6}}$ | $52.30_{2.7}$ | $49.68_8$ | $\mathbf{91.69_{0.5}}$ | $\underline{76.98_{5.8}}$ | $83.68_{7.8}$ |
| CRvNN | $88.58_{12}$ | $51.75_{11}$ | $49.5_3$ | $91.47_{1.2}$ | $\mathbf{77.80_{15}}$ | $\underline{85.38_{3.5}}$ |
| *Beam Tree Models with beam size 5* | | | | | | |
| BT-LSTM | $88.50_{1.1}$ | $50.8_{3.5}$ | $47.14_{17}$ | $90.77_{1.3}$ | $74.25_{7.7}$ | $82.24_{3.5}$ |
| BT-GRC | $88.52_{2.9}$ | $52.32_{4.7}$ | $48.45_{10}$ | $91.29_{1.2}$ | $75.07_{29}$ | $82.86_{23}$ |
| Gumbel-BT-GRC | $88.82_{.9}$ | $51.73_{7.1}$ | $49.67_{1.9}$ | $86.62_{11.3}$ | $72.20_{29}$ | $83.27_{28}$ |
| BT-GRC + Softpath | $88.34_{.7}$ | $51.92_{7.2}$ | $48.01_6$ | $90.86_{9.3}$ | $75.68_{21}$ | $84.77_{11}$ |
| BT-GRC + Gumbelpath | $88.67_{3.4}$ | $51.77_{4.8}$ | $48.15_{16}$ | $88.39_{18}$ | $72_{39}$ | $82.65_{57}$ |
| *Beam Tree Models with beam size 2* | | | | | | |
| BT-GRC | $88.94_6$ | $52.14_{2.8}$ | $\underline{50.08_{7.4}}$ | $91.51_{1.5}$ | $75.21_{23}$ | $82.51_{23}$ |
| BT-GRC + Softpath | $88.52_{4.3}$ | $52.2_{4.4}$ | $47_{2.5}$ | $90.41_{1.8}$ | $75.89_{22}$ | $\mathbf{85.45_{15}}$ |
| BT-GRC + Gumbelpath | $89.27_{3.4}$ | $52.1_4$ | $49.43_{7.2}$ | $83.24_{19}$ | $63.8_{17}$ | $70.42_{4.2}$ |

Table 2: Accuracy on SST2, SST5, and IMDB. Con. refers to Contrast test set, and Count. refers to Counterfactual test set. We report the mean/std of 3. We bold the best resuls and underline the second-best. Subscript represents standard deviation. As an example, $90_1 = 90 \pm 0.1$
.

mysterious. While it struggles on lower beam, it is on par with Softpath with beam size 5. It appears higher beam size is necessary to unlock the potential of Gumbelpath.

## 4.2 LISTOPS ARGUMENT GENERALIZATION RESULTS

**Dataset Settings:** While length generalization (Havrylov et al., 2019; Chowdhury & Caragea, 2021) and depth generalization (Csordás et al., 2022) have been tested before for ListOps, the performance on argument generalization was yet to be considered. In this paper, we also consider what would happen if we increase the number of arguments in the test set beyond the maximum number encountered in the training set. The training set of the original listops data only has $\leq 5$ arguments for each operator. To test for argument generalization we created two new splits - one with 10 arguments per operator and another with 15 arguments per operator. In addition, we also consider the test set of ListOps from Long Range Arena (LRA) dataset (Tay et al., 2021) which serves as check for both length generalization (it has sequences of length 2000) and argument generalization (it has 10 arguments per operators) simultaneously. Results in Table 1.

**Results:** Interestingly, we find all the models perform poorly ($< 90\%$) on argument generalization. However, with the exception of Ordered Memory (OM), BT-cell models (discounting Gumbel-BT-GRC) with higher beam size performs much better than any other models (including otherwise strong contenders like CRvNN). Surprising, OM performs quite well in this split. We do not know the exact reason, but we can eliminate some reasons. First, we know OM is not performing better simply due to better parsing because it even surpasses GoldTreeGRC at times. Second, we also know OM's performance is not just due to a better recursive cell, since its cell (GRC) is shared by many other models that do not perform as well. This may suggest that the memory-augmented RNN style setup in OM is more amenable for argument generalization.

## 4.3 SEMANTIC ANALYSIS (SST AND IMDB) RESULTS

**Dataset Settings:** SST (Socher et al., 2013) and IMDB (Maas et al., 2011) are natural language classification datasets. In the IID splits (standard split) of SST2 and SST5 all the models perform similarly. To better check for OOD performance, we create a length-generalization (LG) split in SST5 (we call this the LG split). Particularly, we only keep sequences of length $\leq 15$ in the training

set, we keep sequences of length $\leq$ 16-29 in the validation set, and we keep all sequences of length $\geq$ 30 in the test set. For IMDB, besides the IID test set, we also test our models on the contrast set from Gardner et al. (2020) and the counterfactual test set from Kaushik et al. (2020). We present our results on these datasets in Table 2.

**Results:** The results in this natural language tasks is rather mixed and do not provide clear consistent differences among these models. There are, however, some interesting highlights. Similar to Shi et al. (2018), we found BalancedTreeGRC can perform competitively in most test splits, however, it performs more poorly than others when it comes to length generalization (SST5 LG split). CRvNN and OM do particularly well in the OOD splits (contrast set and counterfactual split) of IMDB, correlating with their better OOD generalization in synthetic data. BT-GRC + Softpath is also relatively competitive in those splits and better than any other models besides CRvNN and OM. STE Gumbel-based models tend to pareform particularly worse on IMDB.

## 5 ADDITIONAL EXPERIMENTS AND ANALYSIS

**Synthetic Logical Inference**: We present our results on a challenging synthetic logical inference task (Bowman et al., 2015b) in Appendix E.4. We find that most variants of BT-Cell can perform on par with prior SOTA models in this task.

**ListOps Depth Generalization**: We also run experiments to test depth-generalization performance on ListOps (see Appendix E.2)

**Transformers**: We experiment briefly with Neural Data Routers (Csordás et al., 2022) which is a Transformer-based model proven to do well in tasks like ListOps. However, we find that Neural Data Routers (NDRs), despite their careful inductive biases, still struggle with sample efficiency and length generalization compared to strong RvNN-based models. We discuss more in Appendix E.3.

**Natural Language Inference**: We present our results MNLI along with some stress-test splits in Appendix E.5. We find that BT-Cell variants can improve robustness to some stress test splits of MNLI compared to most other models.

**Parse Tree Analysis**: We analyze parsed trees and score distributions in Appendix E.7.

## 6 RELATED WORKS

Goldberg & Elhadad (2010) proposed the easy-first algorithm for dependency parsing. Ma et al. (2013) extended it with beam search for parsing tasks. Choi et al. (2018) integrated easy-first-parsing with an RvNN. Similar to us, Maillard & Clark (2018) used beam search to extend shift-reduce parsing whereas Drozdov et al. (2020) used beam search to extend CYK-based algorithms. However, BT-Cell-based models achieve higher accuracy than the former style of models (eg. BSRP-GRC) and are computationally more efficient than the latter style of models (eg. CYK-GRC) (see Appendix E.6). Similar to us, Collobert et al. (2019) also use beam search in an end-to-end fashion during training but in the context of sequence generation. However, none of the above approaches explored beyond hard top-k operators in beam search. One exception is Xie et al. (2020) where a differentiable top-k operator is used in beam search for language generation (We compare against Xie et al. (2020) in Appendix E.1). We provide an extended related works survey in Appendix F.

## 7 CONCLUSION

Overall, we find all three of Ordered Memory, CRvNN, and BT-Cell are competitive against each other; none being completely superior in all aspects. BT-Cell with Softpath excels in length generalization at ListOps and offers moderate performance on argument generalization but hurts at systematicity (disccussed in Appendix E.4). Ordered Memory excels in argument generalization while struggling in length generalization. CRvNN performs decently in length generalization but struggles in argument generalization. CYK-GRC shows some promise too but is several times more expensive (Appendix E.6) to run while having poor performance in systematicity (Appendix E.4) and argument generalization. We discuss future works in Appendix G.

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

---

**Algorithm 1** Easy First Composition

> **Input:** data $X = [x_1, x_2, ....x_n]$
> **while** True **do**
>     **if** $len(X) == 1$ **then**
>         return $X[0]$
>     **end if**
>     **if** $len(X) == 2$ **then**
>         return $cell(X[0], X[1])$
>     **end if**
>     $Children_L, Children_R \leftarrow X[: len(X) - 1], X[1 :]$
>     $Parents \leftarrow [cell(child_L, child_R) \text{ for } child_L, child_R \text{ in } zip(Children_L, Children_R]$
>     $Scores \leftarrow [scorer(parent) \text{ for } parent \text{ in } Parents]$
>     $index \leftarrow argmax(Scores)$
>     $X[index] \leftarrow Parents[index]$
>     Delete $X[index + 1]$
> **end while**

---

## A    PSEUDOCODES

We present the pseudocode of the easy first composition in Algorithm 1 and the pseudocode of BT-cell in Algorithm 2. Note that the algorithms are written as they are for the sake of illustration: in practice, many of the nested loops are made parallel through batched operations in GPU.

### A.1    BEAM TREE CELL ALGORITHM

Here, we briefly describe the algorithm of BT-cell (Algorithm 2) in words. In BT-Cell, instead of maintaining a single sequence per sample, we maintain some $k$ (initially 1) number of sequences and their corresponding scores (initialized to 0). $k$ is a hyperparameter defining the beam size. Each sequence (henceforth, interchangeably referred to as "beam") is a hypothesis representing a particular sequence of choices of parents. Thus, each beam represents a different path of composition (for visualization see Figure 1). At any moment the score represents the log-probability for its corresponding beam. Now, we describe the steps in each iteration in the recursion of BT-Cell. **Step 1:** similar to gumbel-tree models, we create all candidate parent compositions for each of the $k$ beams. **Step 2:** we score the candidates with the *score* function (defined in §2.2). **Step 3:** we choose top-k highest scoring candidates. We treat the top-k choices as mutually exclusive. Thus, each of the $k$ beams encounters $k$ branching choices, and are updated into $k$ distinct beams (similar to before, the children are replaced by the chosen parent). Thus, we get $k \times k$ beams. **Step 4:** we update the beam scores. The sub-steps involved in the update are described next. **Step 4.1:** we apply a log-softmax to the scores of the latest candidates to put the scores into the log-probability space. **Step 4.2:** we add the log-softmaxed scores of the latest chosen candidate to the existing beam score for the corresponding beam where the candidate is chosen. As a result, we will have $k \times k$ beam scores. **Step 5:** we truncate the $k \times k$ beams and beam scores into $k$ beams and their corresponding $k$ scores to prevent exponential increase of the number of beams. For that, we again simply use a top-k operator to keep only the highest scored beams.

At the end of the recursion, instead of a single item representing the sequence-encoding, we will have $k$ beams of items with their $k$ scores. At this point, to get a single item, we do a weighted summation with the softmaxed scores as the weights as described in §3.

## B    GATED RECURSIVE CELL (GRC)

The Gated Recursive Cell (GRC) was originally introduced by Shen et al. (2019a) drawing inspiration from the Transformer's feed-forward networks. In our implementation, we use the same variant of GRC as was used in Chowdhury & Caragea (2021) where a GELU Hendrycks & Gimpel (2016)

---

**Algorithm 2** Beam Tree Cell

---

**Input:** data $X = [x_1, x_2, ....x_n]$, $k$ (beam size)
$BeamX \leftarrow [X]$
$BeamScores \leftarrow [0]$
**while** True **do**
  **if** $len(BeamX[0]) == 1$ **then**
    $BeamX \leftarrow [beam[0]$ for $beam$ in $BeamX]$
    break
  **end if**
  **if** $len(BeamX[0]) == 2$ **then**
    $BeamX \leftarrow [cell(beam[0], beam[1])$ for $beam$ in $BeamX]$
    break
  **end if**
  $NewBeamX \leftarrow []$
  $NewBeamScores \leftarrow []$

  **for** $Beam, BeamScore$ in $zip(BeamX, BeamScores)$ **do**
    $Parents \leftarrow [cell(beam[i], beam[i+1])$ for $i$ in $range(0, len(beam) - 1)]$
    $Scores \leftarrow log \circ softmax([scorer(parent)$ for $parent$ in $Parents])$
    $Indices \leftarrow topk(Scores, k)$

    **for** $i$ in $range(K)$ **do**
      $newBeam \leftarrow deepcopy(Beam)$
      $newBeam[Indices[i]] \leftarrow Parents[Indices[i]]$
      Delete $newBeam[Indices[i] + 1]$
      $NewBeamX.append(newBeam)$
      $newScore \leftarrow BeamScore + Scores[indices[i]]$
      $newBeamScores.append(newScore)$
    **end for**
  **end for**
  $Indices \leftarrow topk(newBeamScores, k)$
  $BeamScores \leftarrow [newBeamScores[i]$ for $i$ in Indices$]$
  $BeamX \leftarrow [newBeamX[i]$ for $i$ in Indices$]$
**end while**
$BeamScores \leftarrow Softmax(BeamScores)$
Return $sum([score * X$ for $score, X$ in $zip(BeanScores, BeamX)])$

---

activation function was used. We present the equations of GRC here:

$$\begin{bmatrix} z_i \\ h_i \\ c_i \\ u_i \end{bmatrix} = W_2 \, \text{GeLU} \left( W_1^{Cell} \begin{bmatrix} child_{left} \\ child_{right} \end{bmatrix} + b_1 \right) + b_2 \quad (6)$$

$$o_i = LN(\sigma(z_i) \odot child_{left} + \sigma(h_i) \odot child_{right} + \sigma(c_i) \odot u_i) \quad (7)$$

$\sigma$ is $sigmoid$; $o_i$ is the parent composition $\in \mathbb{R}^{d_h \times 1}$; $child_{left}, child_{right} \in \mathbb{R}^{d_h \times 1}$; $W_1^{cell} \in \mathbb{R}^{d_{cell} \times 2 \cdot d_h}$; $b_1 \in \mathbb{R}^{d_{cell} \times 1}$; $W_2 \in \mathbb{R}^{d_h \times d_{cell}}$; $b_1 \in \mathbb{R}^{d_h \times 1}$. We use this same GRC function for any recursive model (including our implementation of Ordered Memory) that constitutes GRC.

## C    BSRP-GRC DETAILS

For the decisions about whether to shift or reduce, we use a scorer function similar to that used in Chowdhury & Caragea (2021). Where Chowdhury & Caragea (2021) use the decision function on concatenation of local hidden states (n-gram window), we use the decision function on the concatenation of last two items in the stack and the next item in the queue. The output is a scalar sigmoid activated logit score $s$. We then treat $log(s)$ as the score for reducing in that step, and $log(1 - s)$

| Model | near-IID | Length Gen. | | | Argument Gen. | | LRA |
|---|---|---|---|---|---|---|---|
| (Lengths) | $\leq 1000$ | 200-300 | 500-600 | 900-1000 | 100-1000 | 100-1000 | 2000 |
| (Arguments) | $\leq 5$ | $\leq 5$ | $\leq 5$ | $\leq 5$ | 10 | 15 | 10 |
| BT-GRC+SOFT | 69 | 44 | 37.1 | 29.4 | 39.5 | 38.6 | 31.6 |

Table 3: Accuracy of BT-GRC+SOFT on ListOps. We report the median of 3 runs. The model was trained on lengths $\leq 100$, depth $\leq 20$, and arguments $\leq 5$.

as the score for shifting in that step. The scores are manipulated appropriately for edge cases (when there are no next item to shift, or when there are no two items in the stack to reduce). Besides that, we use the familiar beam search strategy over standard shift-reduce parsing. Finally the beams of final states are merged through the weighted summation of the states based on the softmaxed scores of each beam similar to BT-Cell models as described in §3.

# D  ST-GUMBEL TOP-K ISSUES

Let us use the notation $argmax_k(.)$ to denote the function for selecting the index of the $k^{th}$ highest value element in a vector or a list. Let us say, we have $s$ as a vector such that $s_i$ represents the element at the $i^{th}$ dimension of $s$. We can now, represent the standard straight-through gumbel softmax Choi et al. (2018) as follows:

$$p = \text{softmax}(s + G) \tag{8}$$

$$(\text{one\_hot}(\text{argmax}_1(p)) - p).detach() + p \tag{9}$$

Here $G$ is the gumbel noise. This strategy allows the forward propagation to have discrete one hot values (one\_hot($\text{argmax}_1(p)$)) while backpropagation to propagate through the soft $p$. However, this creates a discrepancy between forward propagation and backpropagation that leads to biased gradient estimation. This problem can be exacerbated in ST-Gumbel Top-k. In this method, instead of top 1, top $k$ items are selected from $p$. The straight-through estimation involved in the selection of some $r^{th}$ item among the top $k$ selections ($r \leq k$) can be then expressed as:

$$(\text{one\_hot}(\text{argmax}_r(p)) - p).detach() + p \tag{10}$$

Note that $p_{\text{argmax}_r(p)} \leq p_{\text{argmax}_1(p)}$. Thus the discrepancy between $p$ and one\_hot($\text{argmax}_r(p)$) will only increase with increasing $r$. Moreover while originally in ST-Gumbel, when selecting top-1, $p_i$ can be interpreted (in our context) as the probability that $s_i$ is "top-1" item. But in ST-Gumbel Top-k, when selecting the top $r^{th}$, item there is no corresponding interpretation - $p_i$ would still indicate probability for $s_i$ being the "top-1" item not the probability of $s_i$ being the "top $r^{th}$" item. Overall, we attempted ST-Gumbel Top-k more so because it's a simple naive extension of plain top-k but it's not a particularly principled approach. As such, the poor performance is not that surprising. A possible extension could be to transform $p$ to some $p^r$ for every top $r^{th}$ selection such that $p_i^r$ do reflect probability for $s_i$ to be top $r^{th}$ item but that would be non-trivial to do without adding significant overhead (similar to that caused by differentiable sorting - see Appendix E.6).

# E  ADDITIONAL EXPERIMENTS AND ANALYSIS

## E.1  DIFFERENTIABLE SORTING

Besides the three extensions considered for BT-Cell, another strategy can be to simply use differentiable versions of top-k operators or sorting functions (Adams & Zemel, 2011; Grover et al., 2019; Cuturi et al., 2019; Xie et al., 2020; Blondel et al., 2020; Petersen et al., 2021; 2022). However, as we discussed before these methods can lead to issues related to "washing out" of representations due to using soft permutation matrices leading to higher error accumulation (although temperature can be used to partially counteract that (Petersen et al., 2021)). Besides that, using these techniques in a recursive loop can lead to significant overhead and added time complexity. Instead, in this paper, we mainly aim to show that a simple method, namely, softpath can already bring marked improvement compared to plain top-k especially in low beam size settings for listops. We keep a more exhaustive investigation of application of differentiable sorters for beam search as a future work. Nevertheless,

| Model | DG | Length Gen. | | | Argument Gen. | | LRA |
|---|---|---|---|---|---|---|---|
| (Lengths) | $\leq 100$ | 200-300 | 500-600 | 900-1k | 100-1k | 100-1k | 2K |
| (Arguments) | $\leq 5$ | $\leq 5$ | $\leq 5$ | $\leq 5$ | 10 | 15 | $\leq 10$ |
| (Depths) | 8-10 | $\leq 20$ | $\leq 20$ | $\leq 20$ | $\leq 10$ | $\leq 10$ | $\leq 10$ |
| *With gold trees* | | | | | | | |
| GoldTreeGRC | 99.95 | 99.95 | 99.9 | 99.8 | 76.95 | 77.1 | 74.55 |
| *Baselines without gold trees* | | | | | | | |
| CYK-GRC | 99.45 | 99.0 | — | — | 67.8 | 35.15 | — |
| Ordered Memory | **99.95** | 99.8 | 99.25 | 96.4 | **79.95** | **77.55** | **77** |
| CRvNN | 99.9 | 99.4 | 99.45 | 98.9 | 65.7 | 43.4 | 65.1 |
| *Beam Tree Models with beam size 5 (also without gold trees)* | | | | | | | |
| BT-LSTM | 98.9 | 98.5 | 98.1 | 97.7 | 74.75 | 40.75 | 65.05 |
| BT-GRC | **99.95** | **99.95** | **99.95** | **99.9** | 75.35 | 72.05 | 68.1 |
| BT-GRC + Softpath | 99.9 | 99.6 | 98.1 | 97.1 | 78.1 | 71.25 | 75.45 |
| BT-GRC + Gumbelpath | 99.9 | **99.95** | 99.8 | 99.7 | 75.2 | 72.75 | 71.8 |
| *Beam Tree Models with beam size 2 (also without gold trees)* | | | | | | | |
| BT-GRC + Softpath | 96.2 | 95.40 | 93.8 | 91.2 | 64.45 | 51.95 | 51.05 |

Table 4: Accuracy on ListOps-DG. We report the median of 3 runs except in the last block where we report the mean/std of 10 runs as mentioned. Our models were trained on lengths $\leq 100$, depth $\leq 6$, and arguments $\leq 5$. We bold the best results and underline the second-best among models that do not use gold trees.

.

we present some preliminary results here. Xie et al. (2020) used an optimal transport-based differentiable top k method in beam decoding for machine translation. Here, we use their method (SOFT Top-k) and create a new variant of BT-GRC by replacing the softpath with SOFT Top-k. We call this new variant as BT-GRC + SOFT. We run this model in ListOps with the same dataset settings as used in Table 1. We report the results in Table 3. As we can see in Table 3, BT-GRC+SOFT shows very poor performance. This supports our hypothesis that using soft permutation matrix in all recursive iterations may not be ideal because of increase chances of error accumulation and "washing out" through interpolations of what would be distinct path representations. In Appendix E.6, we also demonstrate that using SOFT significantly slows down BT-GRC.

### E.2 LISTOPS-DG EXPERIMENT

**Dataset Settings:** The length generalization experiments in ListOps do not give us an exact perspective in depth generalization[4] capacities. So there is a question of how models will perform in unseen depths. To check for this, we create a new ListOps split which we call "ListOps-DG". For this split, we create $100,000$ training data with arguments $\leq 5$, lengths $\leq 100$, and depths $\leq 6$. We create 2000 development data with arguments $\leq 5$, lengths $\leq 100$, and depths 7. We create 2000 test data with arguments $\leq 5$, lengths $\leq 100$, and depths 8-10. In addition, we also still tested on the same length-generalization splits (which now simultaneously have much higher depths too: $\leq 20$), argument generalization splits, and LRA. The results are presented in Table 4. We only evaluate the models that were promising ($\geq 90\%$ in near IID settings) in the original ListOps split. We report the median of 3 runs for each model (except in the last block of the table).

**Results:** Interestingly, we find that base BT-GRC, CRvNN, and Ordered Memory now does much better in length generalization compared to the original listops split. We think this is because of the increased data (the training data in the original ListOps is $\sim 75000$ after filtering data of length $> 100$ whereas here we generated $100,000$ training data). However, while the median of 3 runs in Ordered Memory is decent, we found one run to have very poor length generalization performance. To investigate more deeply if Ordered Memory has a particular stability issue, we ran Ordered Memory for 10 times with different seeds, and we find that it frequently fails to learn to generalize over length. As a baseline, we also ran BT-GRC similarly for 10 runs and found it to be much more stable. We report the mean and standard deviation of 10 runs of Ordered Memory and BT-GRC

---

[4]By depth, we simply mean the maximum number of nested operators in a given sequence in case of ListOps

| Model | DG | Length Gen. | | | Argument Gen. | | LRA |
|---|---|---|---|---|---|---|---|
| (Lengths) | $\leq 100$ | 200-300 | 500-600 | 900-1k | 100-1k | 100-1k | 2K |
| (Arguments) | $\leq 5$ | $\leq 5$ | $\leq 5$ | $\leq 5$ | 10 | 15 | $\leq 10$ |
| (Depths) | 8-10 | $\leq 20$ | $\leq 20$ | $\leq 20$ | $\leq 10$ | $\leq 10$ | $\leq 10$ |
| *Stability Test: Mean/Std with 10 runs. Beam size 5 for BT-GRC* | | | | | | | |
| Ordered Memory | $99.94_{0.6}$ | $97.58_{32}$ | $78.785_{197}$ | $61.85_{291}$ | $77.66_{30}$ | $69.03_{107}$ | $67.35_{125}$ |
| BT-GRC | $99.84_{1.5}$ | $99.58_{5.8}$ | $98.8_{21}$ | $97.85_{39}$ | $73.82_{57}$ | $66.21_{107}$ | $66.975_{102}$ |

Table 5: Accuracy on ListOps-DG (Stability test). We report the mean and standard deviation of of 10. Our models were trained on lengths $\leq 100$, depth $\leq 6$, and arguments $\leq 5$. Subscript represents standard deviation. As an example, $90_1 = 90 \pm 0.1$

| Model | DG1 | Length Gen. | | | Argument Gen. | | LRA |
|---|---|---|---|---|---|---|---|
| (Lengths) | $\leq 50$ | 200-300 | 500-600 | 900-1000 | 100-1000 | 100-1000 | 2000 |
| (Arguments) | $\leq 5$ | $\leq 5$ | $\leq 5$ | $\leq 5$ | 10 | 15 | $\leq 10$ |
| (Depths) | 8-10 | $\leq 20$ | $\leq 20$ | $\leq 20$ | $\leq 10$ | $\leq 10$ | $\leq 10$ |
| *After Training on ListOps-DG1* | | | | | | | |
| NDR (layer 24) | 96.7 | 48.9 | 32.85 | 22.1 | 65.65 | 64.6 | 42.6 |
| NDR (layer 48) | 91.75 | 34.60 | 24.05 | 19.7 | 54.65 | 52.45 | 39.95 |
| Model | DG2 | Length Gen. | | | Argument Gen. | | LRA |
| (Lengths) | $\leq 100$ | 200-300 | 500-600 | 900-1000 | 100-1000 | 100-1000 | 2000 |
| (Arguments) | $\leq 5$ | $\leq 5$ | $\leq 5$ | $\leq 5$ | 10 | 15 | $\leq 10$ |
| (Depths) | 8-10 | $\leq 20$ | $\leq 20$ | $\leq 20$ | $\leq 10$ | $\leq 10$ | $\leq 10$ |
| *After Training on ListOps-DG2* | | | | | | | |
| NDR (layer 24) | 95.6 | 44.15 | 30.7 | 20.3 | 67.85 | 58.05 | 46 |
| NDR (layer 48) | 92.65 | 38.6 | 29.15 | 22.1 | 73.1 | 64.4 | 50.4 |

Table 6: Accuracy on ListOps-DG1 and ListOps-DG2. We report the max of 3 runs. In ListOps-DG1, NDR was trained on lengths $\leq 50$, depth $\leq 6$, and arguments $\leq 5$. In ListOps-DG1, NDR was trained on lengths $\leq 100$, depth $\leq 6$, and arguments $\leq 5$. Lyaers denote the layers used during inference.

in Table 5. As can be seen, the mean of BT-GRC is much higher than that of Ordered Memory in length generalization splits.

### E.3 NDR Experiments

**Dataset Settings:** Neural Data Routers (NDR) is a Transformer-based model that was shown to perform well in algorithmic tasks including Listops (Csordás et al., 2022). We tried some experiments with it too. We found NDR to be struggling in the original ListOps splits or the ListOps-DG split. We noticed that in the paper (Csordás et al., 2022), NDR was trained in a much larger sample size ($\sim 10$ times more data than in ListOps-DG) and also on lower sequence lengths ($\sim 50$). To better check for the capabilities of NDR, we created two new ListOps split - DG1 and DG2. In DG1, we set the sequence length to 10-50 in training, development, and testing set. We created 1 million data for training, and 2000 data for development and testing. Other parameters (number of arguments, depths etc.) are same as in ListOps-DG split. Split DG2 is the same as ListOps-DG split in terms of data-generation parameters (i.e it includes length sizes $\leq 100$) but with much larger sample size for the training split (again, 1 million samples same as DG1). We present the results in Table 6.

**Results:** We find that even when we focus on the best of 3 runs in the table, although NDR generalizes to slightly higher depths (8-10 from $\leq 6$) (as reported in (Csordás et al., 2022)), it still struggles with splits with orders of magnitude higher depths, lengths, and unseen arguments. Following the suggestions of Csordás et al. (2022), we also increase the number of layers during inference (eg. upto 48) to handle higher depth sequences but that did not help substantially. Thus, even after experiencing more data, NDR generalizes worse than Ordered Memory, CRvNN, or BT-GRC. Moreover, NDR requires some prior estimation of the true computation depth of the task for its hyperparameter setup unlike the other latent-tree models.

| Model | Number of Operations | | | | | |
|---|---|---|---|---|---|---|
| | 8 | 9 | 10 | 11 | 12 | C |
| *With gold trees* | | | | | | |
| GoldTreeGRC | $97.14_1$ | $96.5_2$ | $95.29_{2.5}$ | $94.21_{9.9}$ | $93.67_{7.7}$ | $97.41_{1.6}$ |
| *Baselines without gold trees* | | | | | | |
| Transformer* | 52 | 51 | 51 | 51 | 48 | 51 |
| Universal Transformer* | 52 | 51 | 51 | 51 | 48 | 51 |
| ON-LSTM* | 87 | 85 | 81 | 78 | 75 | 60 |
| Self-IRU† | 95 | 93 | 92 | 90 | 88 | — |
| RecurrentGRC | $93.04_6$ | $90.43_{4.9}$ | $88.48_6$ | $86.57_{5.8}$ | $80.58_{1.5}$ | $83.17_{5.1}$ |
| BalancedTreeGRC | $77.81_{3.6}$ | $72.56_{6.6}$ | $67.54_{6.4}$ | $63.66_{6.6}$ | $57.44_{7.6}$ | $74.45_{10}$ |
| RandomTreeGRC | $86.67_{5.1}$ | $84.78_{8.2}$ | $80.45_{8.2}$ | $76.62_{8.5}$ | $71.71_{4.7}$ | $78.06_{9.7}$ |
| GumbelTreeLSTM | $77.03_{12}$ | $74.62_{6.1}$ | $69.55_{1.7}$ | $67.94_{8.1}$ | $59.95_{10}$ | $78.20_{11}$ |
| GumbelTreeGRC | $93.46_{14}$ | $91.89_{19}$ | $90.33_{22}$ | $88.43_{18}$ | $85.70_{24}$ | $89.34_{29}$ |
| CYK-GRC | $96.62_{2.3}$ | $96.07_{4.6}$ | $94.67_{11}$ | $93.44_{8.8}$ | $92.54_{9.3}$ | $77.08_{27}$ |
| BSRP-GRC | $89.37_{18}$ | $85.92_{30}$ | $83.06_{35}$ | $80.63_{33}$ | $74.91_{42}$ | $81.88_{31}$ |
| CRvNN | $96.9_{3.7}$ | $95.99_{2.8}$ | $94.51_{2.9}$ | $\underline{94.48}_{5.6}$ | $92.73_{15}$ | $89.79_{58}$ |
| Ordered Memory | $\mathbf{97.5}_{\mathbf{1.6}}$ | $\mathbf{96.74}_{\mathbf{1.4}}$ | $94.95_2$ | $93.9_{2.2}$ | $93.36_{6.2}$ | $\mathbf{94.88_7}$ |
| *Beam Tree Models with beam size 5 (also without gold trees)* | | | | | | |
| BT-LSTM | $93.27_{2.5}$ | $92.63_{5.3}$ | $88.55_{10}$ | $87.85_{8.2}$ | $84.56_{12}$ | $73.02_{12}$ |
| BT-GRC | $96.83_1$ | $95.99_{2.4}$ | $95.04_{2.3}$ | $94.29_{3.8}$ | $93.36_{2.4}$ | $\underline{94.17}_{14}$ |
| Gumbel-BT-GRC | $95.56_8$ | $94.14_{11}$ | $92.77_{16}$ | $91.71_{21}$ | $89.84_{18}$ | $86.55_{35}$ |
| BT-GRC + Softpath | $\underline{97.03}_{1.4}$ | $\underline{96.49}_{1.9}$ | $\underline{95.43}_{4.5}$ | $94.21_{6.6}$ | $\underline{93.39}_{1.5}$ | $78.04_{43}$ |
| BT-GRC + Gumbelpath | $96.93_{1.2}$ | $95.82_{1.5}$ | $94.67_{0.6}$ | $93.48_{2.9}$ | $92.58_9$ | $83.63_{47}$ |
| *Beam Tree Models with beam size 2 (also without gold trees)* | | | | | | |
| BT-GRC | $96.63_{2.2}$ | $95.95_{3.7}$ | $\mathbf{95.45}_{\mathbf{3.1}}$ | $93.71_{5.5}$ | $93.28_{3.9}$ | $86.97_{38}$ |
| BT-GRC + Softpath | $96.96_{1.1}$ | $\underline{96.49}_{1.2}$ | $95.38_{1.8}$ | $\mathbf{94.64}_{\mathbf{3.4}}$ | $\mathbf{93.55}_{\mathbf{3.3}}$ | $81.39_{65}$ |
| BT-GRC + Gumbelpath | $96.48_{1.5}$ | $96.02_{4.7}$ | $94.71_{7.2}$ | $93.94_7$ | $92.46_{1.1}$ | $90.87_{50}$ |

Table 7: Mean accuracy and standard deviaton on the Logical Inference for $\geq 8$ number of operations after training on samples with $\leq 6$ operations. We also report results of the systematicity split C. We bold the best results and underline the second-best for all models without gold trees. * indicates that the results were taken from Shen et al. (2019a) and † indicates results from Zhang et al. (2021). Our models were run 3 times on different seeds. Subscript represents standard deviation. As an example, $90_1 = 90 \pm 0.1$

.

### E.4   SYNTHETIC LOGICAL INFERENCE RESULTS

**Dataset Settings:** We also consider the synthetic testbed for detecting logical relations between sequence pairs as provided by Bowman et al. (2015b). Following Tran et al. (2018), we train the models on sequences with $\leq 6$ operators and test on data with greater number of operators (here, we check for cases with $\geq 8$ operators) to check for capacity to generalize to unseen number of operators. Similar to Shen et al. (2019a); Chowdhury & Caragea (2021), we also train the model on the systematicity split $C$. In this split we remove any sequence matching the pattern $*(and(not*))*$ from the training set and put them in the test set to check for systematic generalization.

**Results:** In Table 7, in terms of operation generalization, our proposed BT-Cell models perform similarly to prior SOTA models like Ordered Memory (OM) and CRvNN while approximating GoldTreeGRC for both beam sizes. GRC-based models perform better than comparative LSTM-based models. Unsurprisingly, following discussions in Appendix D, Gumbel-BT-GRC does not perform as well. In terms of systematicity (split C), OM and BT-GRC (with beam size 5) perform similarly (both above $94\%$) and much better than the other models. Lower beam size extensions hurts systematicity performance for BT-GRC which is not too surprising given we are limiting the beam size. Surprisingly, however, softpath/gumbelpath extensions also hurt systematicity. CYK-GRC shows promise in operator generalization but shows poor systematicity as well. Gumbel-GRC

| Model | M | MM | Len M | Len MM | Neg M | Neg MM |
|-------|---|----|----|----|----|----|
| RecurrentGRC | $71.2_3$ | $71.4_4$ | $49_{25}$ | $49.5_{24}$ | $49.3_6$ | $50.1_6$ |
| BalancedTreeGRC | $71.1_5$ | $71.4_1$ | $59_8$ | $60.7_5$ | $50.2_4$ | $50.4_6$ |
| RandomTreeGRC | $72.2_3$ | $72.3_5$ | $61.4_{23}$ | $62.3_{23}$ | $51.7_3$ | $52.7_7$ |
| GumbelTreeGRC | $71.2_7$ | $71.2_6$ | $57.5_{17}$ | $59.6_{12}$ | $50.5_{20}$ | $51.8_{20}$ |
| CRvNN | $72.2_4$ | $72.6_5$ | $62_{44}$ | $63.3_{47}$ | $52.8_6$ | $53.8_4$ |
| Ordered Memory | $72.5_3$ | $\mathbf{73_2}$ | $56.5_{33}$ | $57.1_{31}$ | $50.9_7$ | $51.7_{13}$ |
| *Beam Tree Models with beam size 5* | | | | | | |
| BT-GRC | $71.6_2$ | $72.3_1$ | $64.7_6$ | $66.4_5$ | $\mathbf{53.7_{37}}$ | $\mathbf{54.8_{43}}$ |
| BT-GRC + Softpath | $71.7_1$ | $71.9_2$ | $65.6_{13}$ | $66.7_9$ | $53.2_2$ | $54.2_5$ |
| BT-GRC + Gumbelpath | $72.1_3$ | $71.9_1$ | $66.3_7$ | $66.9_{14}$ | $51.6_{19}$ | $52.2_{21}$ |
| *Beam Tree Models with beam size 2* | | | | | | |
| BT-GRC | $\mathbf{72.6_1}$ | $72.6_2$ | $\mathbf{66.6_5}$ | $\mathbf{68.1_6}$ | $53.3_{21}$ | $54.4_{24}$ |
| BT-GRC + Softpath | $71.1_3$ | $71.9_1$ | $63.7_{17}$ | $65.6_{12}$ | $51.8_{19}$ | $53_{12}$ |
| BT-GRC + Gumbelpath | $67_{45}$ | $67.8_{45}$ | $55.1_{67}$ | $56.2_{67}$ | $47.8_{42}$ | $48.3_{43}$ |

Table 8: Mean accuracy and standard deviaton on MNLI. Our models were run 3 times on different seeds. Subscript represents standard deviation. As an example, $90_1 = 90 \pm 0.1$
.

performs better than fixed tree models (ReccurentGRC or BalancedTreeCell) or RandomTreeCell but still far from SOTA which is not unexpected given its poor results in ListOps.

### E.5 Natural Language Inference Experiments

**Dataset Settings:** We ran our models on MNLI (Williams et al., 2018) which is a natural language inference task. We tested our models on the development set of MNLI and use a randomly sampled subset of $10,000$ data points from the original training set as the development set. Our training setup is different from Chowdhury & Caragea (2021) and other prior latent tree models which combines SNLI (Bowman et al., 2015a) and MNLI training sets (we don't add SNLI data.). We filter sequences $\geq 150$ from the training set for efficiency. We also test our models in various stress tests (Naik et al., 2018). We report the results in Table 8. M denotes matched development set (used as test set) of MNLI. MM denotes mismatched development set (used as test set) of MNLI. LenM denotes length matched stress set from (Naik et al., 2018). LenMM denotes length mismatched stress set from (Naik et al., 2018). NegM denotes negation matched stress set from (Naik et al., 2018). NegMM denotes negation mismatched stress set from (Naik et al., 2018). Length matched/mismatched stress sets add to the length of the premise by adding tautologies. Negation matched/mismatched stress sets add tautologies containing "not" terms which can bias the model to falsely predict contradictions.

**Results:** Results in Table 8 show BT-GRC variants are not particularly better than the other models in the standard matched/mismatched sets. However, discounting Gumbelpath (which tends to break down), even the weakest BT-GRC variants outperform all other models on length matched/mismatched stress test. BT-GRC (with beam 2 or 5) or BT-GRC with Softpath and beam 5 also tend to do marginally better than other models in negation matched/mismatched test sets. Overall most BT-Cell variants show better robustness to stress tests. We ignore testing Gumbel-BT-GRC because it generally is a bad performer.

Overall, given both the performance here and at IMDB, gumbel-based models (including gumbelpath) may be better avoided in general (Also consider that GumbelTreeGRC, here, performs worse than RandomTreeGRC).

### E.6 Efficiency Analysis

**Settings:** In Table 9, we compare the empirical performance of various models in terms of time and memory. We ran each models on ListOps splits of different sequence lengths (200-250, 500-600, and 900-1000). Each split contains 100 samples. We ran each model with the batch size of 1. Other hyperparameters are same as those used for ListOps. Note that we are showing time and memory consumption during training (not inference). All models are ran in an Nvidia RTX A6000 GPU.

| Model | Sequence Lengths | | | | | |
| | $200 - 250$ | | $500 - 600$ | | $900 - 1000$ | |
| | Time | Memory | Time | Memory | Time | Memory |
|---|---|---|---|---|---|---|
| RecurrentGRC | 0.2 min | 0.02 GB | 0.5 min | 0.02 GB | 1.3 min | 0.03 GB |
| BalancedTreeGRC | 0.3 min | 0.02 GB | 1.2 min | 0.03 GB | 2.1 min | 0.04 GB |
| RandomTreeGRC | 0.4 min | 0.33 GB | 1.4 min | 1.86 GB | 3.0 min | 5.18 GB |
| GumbelTreeGRC | 0.5 min | 0.35 GB | 2.1 min | 1.95 GB | 3.5 min | 5.45 GB |
| CYK-GRC | 9.3 min | 32.4 GB | OOM | OOM | OOM | OOM |
| BSRP-GRC | 2.3 min | 0.06 GB | 6.1 min | 0.19 GB | 10.5 min | 0.42 GB |
| Ordered Memory | 8.0 min | 0.09 GB | 20.6 min | 0.21 GB | 38.2 min | 0.35 GB |
| CRvNN | 1.5 min | 1.57 GB | 4.3 min | 12.2 GB | 8.0 min | 42.79 GB |
| *Beam Tree Models with beam size 5* | | | | | | |
| BT-GRC | 1.1 min | 1.71 GB | 2.6 min | 9.82 GB | 5.1 min | 27.27 GB |
| BT-GRC + Softpath | 1.4 min | 2.74 GB | 4.0 min | 15.5 GB | 7.1 min | 42.95 GB |
| BT-GRC + SOFT | 5.1 min | 2.67 GB | 12.6 min | 15.4 GB | 23.1 min | 42.78 GB |
| *Beam Tree Models with beam size 2* | | | | | | |
| BT-GRC | 1.1 min | 0.68 GB | 2.6 min | 3.92 GB | 5.1 min | 10.90 GB |
| BT-GRC + Softpath | 1.4 min | 0.88 GB | 4.0 min | 5.03 GB | 7.1 min | 14.01 GB |

Table 9: Empirical time and memory consumption for various models. Ran on 100 ListOps data of different sequence lengths wiht batch size 1

.

**Discussions:**

Assume $n$ denotes the sequence length, $d$ denotes the hidden state dimensions, $k$ denotes the beam size, and $m$ denotes the number of memory slots (for Ordered Memory).

**RecurrentGRC:** Recurrent models performs reasonably well. While it need to go through $n$ sequential steps but so do most other models. Moreover, computation in each iteration of the sequential loop is relatively simple - it is just the application of the recursive cell function. The space complexity is also very little because of lack of parallel processing accross the sequence length (it has to only store the hidden state memory O($d$)). Thus the memory consumption stays nearly constant with increasing sequence length.

**BalancedTreeGRC:** BalancedTreeGRC is also moderately efficient. It can slightly increase memory consumption because of more parallelized processing. It composes multiple children in every step. However, since it cuts off half of the sequence at every step, its outer sequential loop is only in the order of $O(logn)$. Thus, its memory consumption does not increases as much compared to GumbelTreeGRC or RandomTreeGRC. While the total sequential steps is much less for BalancedTreeGRC, it seems that the added overhead from more parallelized processing per iteration still makes it slightly slower than RecurrentGRC in practice (still it is faster than any other models). However, there may be room for better code optimization for BalancedTreeCell.

**RandomTreeCell and GumbelTreeCell**: Both are similar in terms of complexity. Both chooses one composition at a time requiring them to still go through $n$ seuqntial steps in the outer loop unlike BalancedTreeGRC. However, unlike RecurrentGRC, in each loop, it has to also parallely compute all possible parent compositions - this leads to higher space complexity for each iteration $O(nd^2)$ that also scales up with increasing sequence length $n$ and also higher computational overhead per iteration in the loop. Thus, they run slower than RecurrentGRC and takes much more memory as well. Nevertheless, in comparison to other latent-tree models these are still relatively fast, simple, and lightweight.

**CYK-GRC:** CYK-GRC is the worst offender of all in terms of computational efficiency. Note also that GRC can itself be relatively expensive cell function because of high dimensional feedforward networks – adding up overhead compared to classical CYK models. CYK-GRC takes a chart-filling approach in a dynamic programming style. The number of cells in the filled chart is $n^2/2$. Each cell in the original CYK algorithm can again have multiple options, however, Maillard et al. (2019) simplifies this by keeping only one option per cell through attention-based pooling. Even after that, each cell still has to contain a $d$ dimensional hidden state. This leads to atleast $o(n^2d)$ space

complexity. Moreover, it still has to take $n$ sequential steps in the outer loop to recursively fill up the chart. At the same time each sequential step is also more expensive than any of the other models. For filling up any cell it has to compute all valid possible ways to combine pre-computed cell and then perform an attention pooling. It has to, thus, apply GRC multiple times to all possible ways of composition from prior chart cells. This step can be, however, parallelized (and we do parallelize it). Nevertheless, applying GRC in parellel to all possible left-right children pair (from prior computed chart cells) can still add to memmory consuption and temporal overheads. This reflects in both extremely high memory consumption and also the slowest speed among all else.

**BSRP-GRC:** BSRP-GRC is essentially a form of stack augmented RNN. Similar to Recurrent-GRC, it's memory consuption is low because of limited parallel processing (although its stack size grow with each iteration which reflects in sharply rising memory with increasing sequence length compared to Recurrent GRC). However, it has added expenses in each recurrent iteration due to additional stack operations and shift/reduce decision computation. In effect this makes the model quite slow although better than some others. BSRPC-GRC taking a shift-reduce parsing strategy also need to increase the sequential steps from $n$ to $2n$. Although that doesn't make an asymptotic difference, it still can further contribute to the empirical slowdown. We use a beam size of 5 for BSRP-GRC here for better comparison against BT-GRC.

**Ordered Memory:** Ordered Memory (OM) is also a form of stack augmented RNN and thus, have similar memory advantages as RecurrentGRC (although overall memory consumption is relative increased due to storing multiple slots of hidden states). However, the main bottleneck in OM is time. Precisely, OM utilizes a nested loop. The outerloop is the same $n$ times sequential operation as RecurrentGRC, but in the in each step of that loop, OM has to again sequentially apply GRC over the $m$ memory slots. This leads to a $O(nm)$ sequential steps. Thus, we get a heavy hit to time consumption with OM.

**CRvNN:** The framework of CRvNN is roughly similar to GumbelTreeGRC. However, instead of choosing one parent at a time, it can choose multiple parents at a time, that too in a soft manner. But this comes at a cost. While GumbelTreeGRC can reduce the sequence size with each sequential step, CRvNN has to still maintain the whole sequence in each step with associated "existential probability" for each sequence item. Moreover, computing the composition of contiguous representations also becomes harder. Since in CRvNNs sequence items exists with some probability, what counts as the first existing contiguous item is also probabilistic. In effect, CRvNN needs to create a $n^2$ attention matrix (similar to Transformers but based on existential probabilities instead of dot product attention) to retrieve neighbor (contiguous) elements to create parent candidates. Furthermore, in each sequential steps, this $n^2$ attention matrix needs to be applied multiple times - for example, to get local composition scores for sigmoid modulation and also to retrieve some $w$ local items for its convolution-based decision function. That can also add significant overhead (although still quite tame, in terms of temporal overhead, compared to the memory-slot-wise recursion in OM). Thus, in effect, we get increased memory consumption and time compared to GumbelTreeGRC. At the same time, however, efficiency analysis for CRvNN is complicated by the fact that it can dynamically halt early. That is, it does not need to take full $n$ sequential steps unlike any of the other models (besides BalancedTreeGRC). However, this doesn't mean CRvNN is guaranteed to take $O(logn)$ sequentials steps. Ideally, it is intended to take as many sequential steps as is the induced binary tree depth. Underlying binary tree depths are not necessarily always (or even in average) around $O(logn)$. So while it can give an increase performance boost, there is not a clear boost asymptotically. Moreover, we found it challenging to evaluate CRvNN fairly. In the 100 samples dataset, CRvNN with early halting can just induce bad trees (given lack of enough data to be trained well) and halt very fast (unreflective of realistic performance) but increasing number of samples makes the analysis more cumbersome overall for all other models. So we instead show the "worst case" performance of CRvNN. The "worst case" is the case when CRvNN does not halt early; thus, to simulate the "worst case" we disable early halting. The "worst case" is also still practically relevant and needed to be considered to set up the hyperparameters (like batch size) properly so that the model does not run out of memory during training because of the "worst case" induction. With this setup, we find CRvNN to be in-between in terms of performance. We discuss more about it in comparison to BT-Cell models in the next section.

**BT-GRC:** The time complexity of BT-GRC is $O(n(knd^2 + k^3nd))$. $k$ (beam size) is technically a relative small constant and can be ignored in asymptotic analysis (bringing BT-GRC at a similar complexity level to Gumbel-Tree GRC), but we keep it here for better exposition of the effect of

beam size ($k$). Similar to most other models here, BT-GRC has to take an outer sequential steps of $n$. In each sequential step, similar to GumbelTreeGRC it has to calculate all parent candidates which can lead to a complexity of $O(nd^2)$, but since now we have to do the same for each of the $k$ beams we have a multiplicative effect: $O(knd^2)$. However, this $nd^2$ computation can be done in parallel (all parent candidates are computed in parallel in GumbelTreeGRC). Similarly each beams are independent and can be computed in parallel as well. So in effect the computation cost here is similar to increasing the batch size of GumbelTreeGRC by $k$. The term $k^3nd$ indicates the selection costs of $k$ vectors of size $d$ from $k^2$ vectors of size $d$ for a sequence of size $n$. Again, in practice this can be also parallelized by creating a $k^2$ permutation matrix and performing a single matrix multiplication in CUDA. Overall, this doesn't add as much cost over parent composition.

As we a priori suspected, empirically, we also verified the increased time/memory cost of BT-GRC in different sequence length to match the effect of increasing the batch size of GumbelTreeGRC by $k$. Even though, ultimately, $k$ is a constant, in practice, this can still lead to a significant expense. This is where small $k$ (example beam 2) can be valuable. As we can see because most computation through $k$ is parallelized the time is not changed as much in changing from beam size ($k$) 2 to 5. But the memory consumption can be significantly decreased with lower beam ($k = 2$). Thus, BT-Cell with beam size 2 can be an attractive choice here particularly when it can still perform relatively on par with bigger beam models on synthetic logical inference and natural language tasks. Also with softpath and beam size 2, listops performance is still decent.

Another interesting point to note is that BT-GRC does not rely on any $n^2$ matrix as CRvNN. Thus, we can see that its memory consumption scales better with increasing length than CRvNN. For example, in sequence length 200-250, CRvNN took slightly less memory than BT-GRC, but in sequence length 900-1000, CRvNN takes nearly twice as much memory compared to BT-GRC.

However, we wouldn't claim that CRvNN is strictly worse than BT-GRC in efficiency departments because CRvNN can be made more efficient by bounding its outer sequential loop and dynamic halting (but these factors are more tricky to fairly analyze).

**BT-GRC + Softpath:** Although the forward propagation complexity should be nearly the same for Softpath, in practice we find that adding Softpath increases both the memory and time significantly compared to base BT-GRC. We find that this is because of added backpropagation expenses because of more complicated gradient propagation. We verified this by keep the forward network of Softpath variant the same and using Pytorch's $detach()$ function to cut the gradient from Top-K Softpath selection. This change leads to similar empirical efficiency to base BT-GRC. Nevertheless, despite the added costs, Softpath is still comparable to CRvNN and much faster than OM, CYk-GRC, SOFT top-k, and BSRP-GRC. Moreover, Softpath with beam size 2 still remains an attractive option with further lowered memory consumption.

**BT-GRC + SOFT:** As we already claimed before using differentiable sorting algorithm (SOFT Top-k) (Xie et al., 2020) in a recursive loop can bring significant overhead and slowdown. We show it empirically in Table 9. Replacing Softpath with SOFT Top-k can increase the time taken by around $3\times$ compared to BT-GRC+Softpath.

### E.7   PARSE TREE ANALYSIS

In this section, we analyze the induced structures of BT-Cell models. Note, however, although induced structures can provide some insights to the model, we can draw limited conclusions from them. First, if we take a stance similar to Choi et al. (2018) in considering it suitable to allow different kinds of structures to be induced as appropriate for a specific task then it's not clear how structures should be evaluated by themselves (besides just the donwstream task evaluations). Second, the extracted structures may not completely reflect what the models may implicitly induce because the recursive cell can override some of the parser decisions (given how there is evidence that even simple RNNs Bowman et al. (2015b) can implicitly model different tree structures within its hidden states to an extent even when its explicit structure always conform to the left-to-right order of composition). Third, even if the extracted structure perfectly reflects what the model induces, another side of the story is the recursive cell itself and how it utilizes the structure for language understanding. This part of the story can still remain unclear because of the blackbox-nature of neural nets. Nevertheless, extractive structures may still provide some rough idea of the inner workings of BT-Cell variants.

| Score | Parsed Structures |
|---|---|
| | **BT-GRC (beam size 5)** |
| 0.42 | ((i (did not)) (((like a) (single minute)) ((of this) film))) |
| 0.40 | (((i (did not)) ((like a) (single minute))) ((of this) film)) |
| 0.20 | ((i (did not)) (((like a) ((single minute) of)) (this film))) |
| 0.40 | ((i (shot an)) ((elephant in) (my pajamas))) |
| 0.21 | (((i shot) (an elephant)) ((in my) pajamas)) |
| 0.19 | (((i shot) (an elephant)) (in (my pajamas))) |
| 0.19 | ((i shot) ((an elephant) ((in my) pajamas))) |
| 0.40 | ((john saw) ((a man) (with binoculars))) |
| 0.40 | (((john saw) (a man)) (with binoculars)) |
| 0.20 | ((john (saw a)) ((man with) binoculars)) |
| 0.61 | (((roger (dodger is)) (one (of the))) (((most compelling) (variations of)) (this theme))) |
| 0.40 | (((roger (dodger is)) ((one (of the)) (most compelling))) ((variations of) (this theme))) |
| | **BT-GRC (beam size 2)** |
| 0.50 | ((i ((did not) like)) (((a single) minute) ((of this) film))) |
| 0.50 | ((i (((did not) like) (a single))) ((minute of) (this film))) |
| 0.50 | ((i (shot an)) ((elephant in) (my pajamas))) |
| 0.50 | ((i ((shot an) elephant)) ((in my) pajamas)) |
| 0, 51 | ((john (saw a)) ((man with) binoculars)) |
| 0.49 | (john (((saw a) man) (with binoculars))) |
| 1.0 | ((roger ((dodger is) one)) ((((of the) most) (compelling variations)) ((of this) theme))) |

Table 10: Parsed Structures of BT-GRC trained on MNLI. Each block represents different beams.
.

In Table 10, we show the parsed structures of some iconic sentences by BT-GRC after it is trained on MNLI. We report all beams and their corresponding scores. Note, although beam search ensures that the sequence of parsing actions for each beam is unique, different sequences of parsing action can still lead to the same structure. Thus, some beams end up being duplicates. In such cases, for the sake of more concise presentation, we collapse the duplicates into a single beam and add up their corresponding scores. This is why we can note in Table 10 that we sometimes have fewer induced structures than the beam size.

At a rough glance, we can see that the different induced structures roughly correspond to human intuitions. One interesting appeal for beam search is that it can more explicitly account for ambiguous interpretations corresponding to ambiguous structures. For example, *"i shot an elephant in my pajamas"* is ambiguous with respect to whether it is the elephant who is in the shooter's pajamas, or if it is the shooter who is in the pajamas. The induced structure (beam size 5 model in Table 10) *(((i shot) (an elephant)) ((in my) pajamas))* corresponds better to the latter interpretation whereas *((i shot) ((an elephant) ((in my) pajamas)))* corresponds better to the former interpretation (because "an elephant" is first composed with "in my pajamas").

Similar to above, *"john saw a man with binoculars"* is also ambiguous. Its interpretation is ambiguous with respect to whether it is John who is seeing through binoculars, or whether it is the man who just possesses the binoculars. Here, again, we can find (beam size 5 model in Table 10) that the induced structure *(((john saw) (a man)) (with binoculars)* corresponds better to the former interpretation whereas *((john saw) ((a man) (with binoculars)))* corresponds better to the latter. Generally, we find the score distributions to have a high entropy. A future consideration would be whether we should add an auxiliary objective to minimize entropy.

In Table 11, we show the parsed structures of the same sentences by BT-GRC+Softpath after it is trained on MNLI, and in Table 12, we show the same for BT-GRC+Gumbelpath. Most of the points above applies here for Softpath and Gumbelpath as well. Interestingly, Softpath and Gumbelpath seemed to have a relatively lower entropy distribution - that is most evident in beam size 2. We also note that Gumbelpath structures are of a similar quality despite having much lower empirical performance in MNLI.

| Score | Parsed Structures |
|---|---|
| **BT-GRC + Softpath (beam size 5)** | |
| 0.42 | (((i did) (not like)) (((a single) minute) ((of this) film))) |
| 0.20 | ((((i did) not) ((like a) single)) ((minute of) (this film))) |
| 0.19 | ((((i did) (not like)) ((a single) minute)) ((of this) film)) |
| 0.19 | (((i (did not)) ((like a) single)) ((minute of) (this film))) |
| 0.41 | (((i shot) an) ((elephant in) (my pajamas))) |
| 0.21 | (((i shot) (an elephant)) ((in my) pajamas)) |
| 0.19 | ((i (shot an)) ((elephant in) (my pajamas))) |
| 0.19 | ((((i shot) an) (elephant in)) (my pajamas)) |
| 0.21 | ((john (saw a)) ((man with) binoculars)) |
| 0.20 | (((john saw) (a man)) (with binoculars)) |
| 0.20 | ((john saw) ((a man) (with binoculars))) |
| 0.19 | ((john ((saw a) man)) (with binoculars)) |
| 0.40 | (((roger dodger) (is one)) ((((of the) most) (compelling variations)) ((of this) theme))) |
| 0.21 | (((roger (dodger is)) ((one of) the)) (((most compelling) variations) ((of this) theme))) |
| 0.20 | ((((roger dodger) (is one)) ((of the) most)) ((compelling variations) ((of this) theme))) |
| 0.19 | ((roger (dodger is)) ((((one of) the) ((most compelling) variations)) ((of this) theme))) |
| **BT-GRC + Softpath (beam size 2)** | |
| 0.57 | ((i ((did not) like)) (((a single) minute) ((of this) film))) |
| 0.43 | ((i ((did not) like)) (((a single) (minute of)) (this film))) |
| 0.54 | ((i ((shot an) elephant)) ((in my) pajamas)) |
| 0.46 | ((i (shot an)) ((elephant in) (my pajamas))) |
| 0.55 | ((john (saw a)) ((man with) binoculars)) |
| 0.45 | ((john ((saw a) man)) (with binoculars)) |
| 0.53 | ((roger ((dodger is) one)) ((((of the) most) (compelling variations)) ((of this) theme))) |
| 0.47 | (((roger ((dodger is) one)) ((of the) most)) ((compelling variations) ((of this) theme))) |

Table 11: Parsed Structures of BT-GRC + Softpath trained on MNLI. Each block represents different beams.

.

We found the structures induced by BT-Cell variants after training on SST5 or IMDB to be more ill-formed. This may indicate that sentiment classification does not provide a strong enough signal for parsing or rather, exact induction of structures are not as necessary (Iyyer et al., 2015). We show the parsings of these models after training on IMDB and SST datasets in a text file included in the supplementary.

## F  EXTENDED RELATED WORKS

Initially RvNN Pollack (1990); Socher et al. (2010) was used with user-annotated tree-structured data. Some explored use of heuristic trees such as balanced trees for RvNN-like settings (Munkhdalai & Yu, 2017; Shi et al., 2018). In due time, several approaches were introduced for dynamically inducing structures from data for RvNN-style processing. This includes the greedy easy-first framework using children-reconstruction loss (Socher et al., 2011) or gumbel softmax (Choi et al., 2018), RL-based frameworks (Havrylov et al., 2019), CYK-based framework (Le & Zuidema, 2015; Maillard et al., 2019; Drozdov et al., 2019; Hu et al., 2021), shift-reduce parsing or memory-augmented or stack-augmented RNN frameworks (Grefenstette et al., 2015; Bowman et al., 2016; Yogatama et al., 2017; Maillard & Clark, 2018; Shen et al., 2019a; DuSell & Chiang, 2020; 2022), and soft-recursion-based frameworks (Chowdhury & Caragea, 2021; Zhang et al., 2021). Besides RvNNs, other approaches range from adding information-ordering biases to hidden states in RNNs (Shen et al., 2019b) or even adding additional structural or recursive constraints to Transformers (Wang et al., 2019; Nguyen et al., 2020; Fei et al., 2020; Shen et al., 2021; Csordás et al., 2022).

| Score | Parsed Structures |
|-------|-------------------|
| **BT-GRC + Gumbelpath (beam size 5)** | |
| 0.42 | (((i (did not)) (like a)) ((single (minute of)) (this film))) |
| 0.38 | (((i did) ((not like) (a single))) ((minute (of this)) film)) |
| 0.20 | (((i did) ((not like) (a single))) ((minute (of this)) film)) |
| 0.43 | ((i (shot an)) ((elephant in) (my pajamas))) |
| 0.20 | (((i shot) (an elephant)) ((in my) pajamas)) |
| 0.19 | (((i shot) (an elephant in)) (my pajamas)) |
| 0.18 | (((i (shot an)) (elephant in)) (my pajamas)) |
| 0.39 | (((john saw) (a man)) (with binoculars)) |
| 0.39 | ((john saw) ((a man) (with binoculars))) |
| 0.21 | ((john (saw a)) ((man with) binoculars)) |
| 0.41 | (((roger (dodger is)) (one (of the))) ((most (compelling variations)) ((of this) theme))) |
| 0.21 | (((roger dodger) (is one)) (((of the) (most (compelling variations))) ((of this) theme))) |
| 0.20 | (((roger dodger) ((is one) (of the))) ((most (compelling variations)) ((of this) theme))) |
| 0.19 | (((roger (dodger is)) ((one (of the)) (most (compelling variations)))) ((of this) theme)) |
| **BT-GRC + Gumbelpath (beam size 2)** | |
| 0.58 | (((((i did) (not like)) (a single)) (minute (of this))) film) |
| 0.42 | (((((i did) (not like)) (a single)) (minute of)) (this film)) |
| 0.64 | (((i (shot an)) (elephant (in my))) pajamas) |
| 0.36 | (((i (shot an)) (elephant in)) (my pajamas)) |
| 0.53 | ((john (saw a)) (man (with binoculars))) |
| 0.47 | (((john (saw a)) (man with)) binoculars) |
| 0.51 | (((((roger dodger) (is one)) (of the)) (most (compelling variations))) (of (this theme))) |
| 0.49 | ((((((roger dodger) (is one)) (of the)) (most (compelling variations))) (of this)) theme) |

Table 12: Parsed Structures of BT-GRC + Gumbelpath trained on MNLI. Each block represents different beams.

.

# G  LIMITATIONS AND FUTURE WORKS

The ideal future direction would be to extend to methods which generalize better in all aspects while maintaining computational efficiency and at the same time having a more flexible architecture for handling more free structures like non-projective trees or directed acyclic graphs and also richer classes of languages (DuSell & Chiang, 2022; Del'etang et al., 2022). Another direction to explore would be to linearize the recursion particularly taking inspiration from state space models (Gu et al., 2022) to make these models more scalable.

# H  HYPERPARAMETERS

For all recursive/recurrent models, we use a linear layer followed by layer normalization for initial leaf transformation before starting the recursive loop (similar to Shen et al. (2019a); Chowdhury & Caragea (2021)). Overall we use the same boilerplate classifier architecture for classification and the same boilerplate sentnece-pair siamese architecture for logical inference as Chowdhury & Caragea (2021) over our different encoders. In practice, for BT-Cell, we use a stochastic top-k through gumbel perturbation similar to Kool et al. (2019). However, we find deterministic selection to work similarly.

In terms of the optimizer, hidden size, and other hyperparameters besides dropout, we use the same ones as used by (Chowdhury & Caragea, 2021) for all models for corresponding datasets; for number of memory slots and other ordered memory specific parameters we use the same ones as used by (Shen et al., 2019a). For BSRP-Cell we use a beam size of $8$ (we also tried with $5$ but results were similar or slightly worse). We use a dropout rate of $0.1$ for logical inference for all models (tuned on the validation set using grid search among $[0.1, 0.2, 0.3, 0.4]$ with 5 epochs per run using BalancedTreeCell for GRC-based models and GumbelTreeLSTM for LSTM based models). We

use dropouts in the same places as used in (Chowdhury & Caragea, 2021). We then use the same chosen dropouts for ListOps. We tune the dropouts for SST in the same way (but with a maximum epoch of 20 per trial) on SST5 using RecurrentGRC for GRC-models, and Gumbel-Tree-LSTM for LSTM models. After tuning, for GRC-based models in SST5, we found a dropout rate of $0.4$ for input/output dropout layers, and $0.2$ for the dropout layer in the cell function. We found a dropout of $0.3$ for LSTM-based models in SST5. and We share the hyperparameters of SST5 with SST2 and IMDB. For MNLI, we used similar settings as Chowdhury & Caragea (2021).

For NDR experiments, we use the same hyperparameters as used for ListOps by Csordás et al. (2022). The hyperparameters will also be available in code.

All codes are run in a single RTX A6000 GPU.

