# OpenReview forum: "Beam Tree Recursive Cells"
_ICLR.cc/2023/Conference — Submitted to ICLR 2023_

### Official Review · Reviewer_dDzD · 2022-10-24

**Confidence:** 3
**Correctness:** 3
**Technical Novelty And Significance:** 2
**Empirical Novelty And Significance:** 2
**Recommendation:** 6

**Clarity, Quality, Novelty And Reproducibility:**

* Clarity: the paper is easy to read, but relation between BT-cell and CYK can be made clearer. The difference between them is: CYK approach *fuses* subtree representations; whereas BT-cell keeps them separated and only *fuses* them in the end. Seeing this way, BT-cell  has a strong similarity to CYK with (1) top-k pruning, and (2) replace pooling.

* Quality: the quality of the paper can be improved with experiments on hyper-param choices, analyses on the impact of beam mechanism, and the impact of different components (beam size, top-k operator)

* Originality: the work is quite incremental. There are several ways to see how the work related to existing methods in the literature: for easy-first parsing, the work replaces argmax by top-k. The work can also be seen as a restriction of CYK with "beam"-pooling.

**Strength And Weaknesses:**

The BT-cell is a straightforward extension for argmax-based easy-first parsing and thus it is clear and easy to understand. The paper however does't explain why this extension is helpful.

Firstly, the paper lacks analyses about the impact of beam search and found structures among the beams. For instance, by examining the beams, can we find *good* structures that support required compositionality? What if we vary the beam size? What are distributions of scores? Also, as the paper claims that top-k is good enough although it is non-differentiable, would there be an analysis looking into the beams to support the argument given right below eq (3)?

The experiment setting is pretty unfair to the baselines. The paper compares several variations of BT-cells (with different beam sizes, top-k operators...) against the baselines. But that is not much different from fine-tuning on test sets, where hyper-params are beam size and top-k operator. For instance, in Tab1 we can see that BT-GRC performs very well on 'C', with '+softpath' it does better on '7,8,...,12' but much worse on 'C'. BT-cell with different beam sizes also yields different results ( tab1 vs tab3). However in the end, all the conclusions are for BT-cell in general, rather than some specific BT-cell configuration.

The paper claims that CYK approaches are expensive, but there's no complexity analysis for the proposed BT-cell.





**Summary Of The Paper:**

The paper proposes a new method, called Beam Tree Recursive Cells, for sentence representation recursively using beam search. The BT-cell extracts a beam of parses (using the same mechanism of easy-first parsing, replacing argmax with top-k) when processing a sentence, and then combining all beams at the end for a sentence representation. Although top-k is non-differentiable, the paper shows that it is quite effective enough for training, without the need for back-prop though top-k. The paper demonstrates that BT-cell is effective for artificial tasks Logical Inference, ListOps, and real tasks SST and IMDB.

**Summary Of The Review:**

The paper doesn't meet the acceptance standard:
* its quality should be improved so that the the choices and their impacts are understood better
* the work is quite incremental, i.e. the contribution doesn't seem significant.

---

> ### Author Response · Authors · 2022-11-18
> **Response 4.1**
>
> Thank you for your great suggestions!
>
> > The paper however does't explain why this extension is helpful.
>
> Thank you for pointing out the need for a better presentation.
>
> We have added more motivation in Section 3:
>
> “Gumbel-Tree models, as described, are relatively fast and simple but they are fundamentally based on a greedy algorithm for a task where the greedy solution is not guaranteed to be optimal. On the other hand, adaptation of dynamic programming-based CYK-models (Maillard et al. 2019) leads to high computational complexity (discussed more in Appendix E.6). A “middle way” between the two extremes is then to simply extend Gumbel-Tree models with beam-search to make it less greedy while still being less costly than CYK-parsers (See Appendix E.6). Moreover, using beam-search also provides additional opportunity to recover from local errors whereas a greedy single-path approach (like Gumbel Tree models) will be stuck with any errors made. All these factors motivate the framework of Beam Tree Cells (BT-Cell).”
>
> > Firstly, the paper lacks analyses about the impact of beam search and found structures among the beams. For instance, by examining the beams, can we find _good_ structures that support required compositionality? What if we vary the beam size? What are distributions of scores? Also, as the paper claims that top-k is good enough although it is non-differentiable, would there be an analysis looking into the beams to support the argument given right below eq (3)?
>
> We have added extra analysis on parse structures and score distributions in Appendix E7.
>
> Note that the impact of beam size and top-k operators on overall evaluation (if not parse trees) of different tasks were already present.
>
> Note that for our purpose, we consider "good" structure as whatever that helps with the task (similar stance to Choi et al. 2018) instead of treating some structure as “intrinsically good". Moreover, below eqn 3 (in pre-revised version), we merely said (quoting the pre-revised version): "We believe that as a combination of these two factors, plain BT-Cell even with non-differentiable top-k operators can perform well for structure-sensitive tasks".  That it “performs well” is supported empirically by how BT-GRC, even with plain top-k, can get near $90\\%$ in logical inference and ListOps. Also we note a massive boost is achieved just from making the simple beam search extension to Gumbel Tree GRC all else kept the same in ListOps and Logical Inference (same holds when comparing Gumbel Tree LSTM vs BT-LSTM). Ablations like RecurrentGRC, BalancedTreeGRC, RandomTreeGRC show that such boost cannot be achieved by heuristic-structures. This shows that BT-GRC can learn task-relevant structures.
>
>
> > The paper claims that CYK approaches are expensive, but there's no complexity analysis for the proposed BT-cell.
>
> We have added complexity analysis in Appendix E6. Thank you for the suggestion.
>
> > the paper is easy to read, but relation between BT-cell and CYK can be made clearer. The difference between them is: CYK approach _fuses_ subtree representations; whereas BT-cell keeps them separated and only _fuses_ them in the end. Seeing this way, BT-cell has a strong similarity to CYK with (1) top-k pruning, and (2) replace pooling.
>
> We don't believe there is particularly any relation between BT-Cell and CYK beyond that both approaches attempt to simulate RvNNs.
> BT-Cell extends a greedy algorithm with beam search. CYK in practice (like Maillard et al. 2019) is  a dynamic programming algorithm with vector-averaging within cells for tractability. If we use top-k on CYK it would require K separate dynamic-programming charts. One chart is already plenty expensive memory-wise. BT-Cell maintains k sequences rather than charts. (See Appendix E6 for more discussions)
>
> > the quality of the paper can be improved with experiments on hyper-param choices, analyses on the impact of beam mechanism, and the impact of different components (beam size, top-k operator)
>
> Results for different top-k, and different beam sizes were already provided for all datasets. We have added analysis of parses for different beam sizes and top-k in Appendix E7.
>
> > -   Originality: the work is quite incremental. There are several ways to see how the work related to existing methods in the literature: for easy-first parsing, the work replaces argmax by top-k. The work can also be seen as a restriction of CYK with "beam"-pooling.
>
> As we argued above, BT-Cell is not a “beam”-pooling restriction of CYK.
>
> Also please see our General Responses - particularly, the section on "BT-Cell novelty” (General Response 1) and "Softpath novelty” (General Response 2). We would also like to highlight our empirical contributions (see ``Empirical Contributions” in our General Response 2).

---

> > ### Comment · Reviewer_dDzD · 2022-11-23
> > **reply**
> >
> > I would like to thank the authors for rich information.
> >
> > Re the relationship between BT-Cell and CYK, sorry that I didn't express my thought well. Imagine that we unroll the beam-search of BT-Cell on a chart. Intuitively, we maintain a priority queue of size k to keep k best scores. In each cell (i,j) of the chart, we will look for all possible combinations of two lower cells (i,k) and (k+1,j). Each of these two charts contain *less* than k subtrees (possibly 0 subtrees). We now prune all the combinations in cell (i,j) and keep only those whose scores, after pushed into the queue, are still kept in the queue. In general, each cell contains less than k vectors, and the pooling operator is here is a pruner with a global priority queue of size k.
> >
> > I agree that the authors didn't over-claim the power of BT-Cell as I criticized for 'hyper-parameter tuning', though I don't agree with the authors '[e]ither way, we can also say that RandomTreeGRC, BalancedTreeGRC, RecurrentTreeGRC, GumbelTreeGRC, GumbelTreeLSTM etc. are all variations of the same framework - just switching the scoring function.' That is because k is already a hyper-param in BT-Cell whereas switching the scoring function is not part of the other models. The question here is why not search for k (and thus a model variant) on a dev set? I believe that any analyses (including ablation test) on test sets should be avoided because test sets should always be "unseen".
> >
> > I like E6 (and especially thank the authors for that). I suggest to put table 9 in the main text because it really supports the motivation in section 3.
> >
> > In general, I raise my score from 5 to 6. What I'm not satisfied is the way showing model variants on test sets. I suggest selecting model variants on dev set only. If the authors want to show that k=2 is good enough, please write a separated section arguing that. I also don't see an analysis on varying k=2,3,4,5,... (I see why k > 5 is troublesome due to memory limit).

---

> > > ### Author Response · Authors · 2022-11-26
> > > **Round 2 Response 4.1 Part 1/2**
> > >
> > > Thank you for the response, suggestions, and clarifications!
> > >
> > > **BT-Cell vs CYK**
> > >
> > > Thank you for the clarification. We have a couple of points in response to this idea:
> > > 1. While the operations of BT-Cell could be visualized as unrolling in a chart it doesn’t immediately become a variation of CYK simply by that reason. Particularly, given that we are now possibly having empty chart cells, and cells with different numbers of explicit spans and such -- it already becomes a very different algorithm from the original CYK and its standard adaptations.
> > > 2. Perhaps, however, there is a way we can see BT-Cell as CYK+beam-search coupled with some form of “intelligent pruning” over the chart.  Even then we think the exact connections are not obvious. For instance, choice of composition between two cells in a lower level in your suggested formalism happens at the level of contiguous spans whereas beam selections in BT-Cell happen at the level of whole (intermediate) sequences. For example, in BT-Cell, we may have to consider between choices like c1: (x1, (x2,x3), x4, (x5,x6)) and c2: ((x1,x2),x3,(x4,x5),x6). So when trying to compose the (x1,x2,x3), it’s not just a choice between (x1,(x2,x3)) and ((x1,x2),x3) from lower level cells having [(x2,x3)], [x1] and [(x1,x2)], [x3] ([] denoting cell boundaries in a chart)  but a choice between c1 and c2 where choosing c1 is entangled with  maintaining the choice (x5,x6) and choosing c2 is entangled with maintaining the choice (x4,x5). It’s unclear how a rigorous connection can be made by your formalism such that these nuances are preserved without making too many modifications to CYK (with unrestricted modifications any algorithm can be shown to be any other algorithm especially given any RvNN algorithms considered in the scope of the paper  is surfing in the space of projective tree structures).
> > > 3. We think the broader insight here is that both CYK and BT-Cell can maintain different mutually exclusive choices simultaneously. This is partly acknowledged in the current version by framing BT-Cell as a "middle way" between Greedy Gumbel-Tree-Cell and CYK in the motivation (Section 3).
> > > 4. Your idea is independently interesting and it may be possible to draw on it to develop different algorithms for adapting CYK that do span level choices combining ideas from beam search. Probably something interesting to explore in future works.

---

> > > ### Author Response · Authors · 2022-11-26
> > > **Round 2 Response 4.1 Part 2/2**
> > >
> > >
> > > **On the issue of “Hyperparameter Tuning” on Test Set as an accidental side-effect of showing results of different beam sizes on test set (even though all hyperparameters were tuned on development set in the standard sense):**
> > >
> > >
> > > We agree that  k is closer to a hyperparameter whereas GRC/LSTM/Scorer-function can be considered closer to model variants. However, these distinctions can be blurry and even you originally treated not just k but also the top-k operators (STE-Gumbel Top-K, Gumbelpath, Softpath, Plain Top-k) as not model variants but as akin to hyperparameters that add unfairness: “The paper compares several variations of BT-cells (with different beam sizes, top-k operators...) against the baselines.”. In fact a stronger emphasis was given in your original comment on the variation of top-k operations. That is why we pointed out that analogously we can think of GumbelTreeGRC/RandomTreeGRC/others as simply variants of the score function “hyperparameter”.
> > >
> > > Given that now your emphasis is on the beam size (k), we take it that you agree that it’s hard to treat top-k operators themselves (if not beam size k) as hyperparameters without also treating the score-functions in Gumbel-TreeGRC/Random-TreeGRC as hyperparameters in a more generalized model-framework?
> > >
> > > Regarding the hyperparameter tuning issue with k, we acknowledge your concern but we want to make a few points:
> > >
> > > Points:
> > >
> > > * Note that if we are treating only beam size k as the hyperparameter, given that we only test two settings of k (5 and 2) (out of an exponential space), at the very least we believe the severity of the “hyperparameter tuning on test set” concern is mitigated. Moreover, we believe the “unfairness” objection does not apply as much either given that with proper top-k operators (softpath/plain top-k), generally both k=2 and k=5 performs better than most of the baselines or both k perform comparably enough such that similar conclusions apply for both.
> > > * While ideally perhaps we should test “ablations” in dev sets (and we will keep it in mind henceforth), standardly that’s rarely done in the community. Countless examples of published papers can be given (upon request)  where ablations are done in test sets. Showing choices on different hyperparameters is also a standard practice in the name of analysis.
> > >
> > > Trade offs to consider:
> > > * Also note as we stated we show a different sized k (k=2) for a reason - i.e to show how much we can get away with a minimally costly version of beam search. Simply treating it as a hyperparameter and not showing its results will blind us from this insight.
> > > * One way to resolve this would be to show the results of k=2 only in the validation set as you suggested but again that would rob us from the insights from how k=2 performs in OOD sets since the current validation sets are not split based on different OOD factors. To address this issue,  we would have to create new validation sets corresponding to the test OOD distributions but overall it may also make the presentation more confusing and it cannot be easily done for OOD test sets in natural language tasks where we don’t have control over the data generating process.
> > >
> > > However, if the quality of the paper would be better if it only shows validation results for k=2 even if all OOD results with k=2 are deleted or if only partial OOD results with k=2 are shown where constructing a corresponding OOD validation set is possible, then we can attempt to make those corresponding changes in the next version of the paper.
> > >
> > > The above change would be mostly presentational (elimination of some k=2 OOD results, inclusion of k=2,k=5 validation results) and even if we completely ignore k=2 results altogether the conclusions and main points of the paper still stand roughly unchanged.
> > >
> > > **More Beam Sizes**
> > >
> > > We will consider some analyses with more beam sizes for the next version. However note that:
> > >
> > > * Analyzing results on different beam sizes on test sets can exacerbate your own concern but analyzing them on the existing validation set would again rob us insights from OOD performance.
> > > * If we are talking about exclusively parse-analyses, we didn’t find any particularly surprising insights between parses in k=2 and k=5 in Appendix E7. So it’s unlikely there would be anything remarkable to find in between the extremes of k=2 and k=5 besides more of the same.

---

> ### Author Response · Authors · 2022-11-18
> **Response 4.2**
>
>
> > The experiment setting is pretty unfair to the baselines. The paper compares several variations of BT-cells (with different beam sizes, top-k operators...) against the baselines. But that is not much different from fine-tuning on test sets, where hyper-params are beam size and top-k operator. For instance, in Tab1 we can see that BT-GRC performs very well on 'C', with '+softpath' it does better on '7,8,...,12' but much worse on 'C'. BT-cell with different beam sizes also yields different results ( tab1 vs tab3). However in the end, all the conclusions are for BT-cell in general, rather than some specific BT-cell configuration.
>
> The final conclusions are not for BT-Cell in general. Quoting from the pre-revised version's conclusion:
> "BT-Cell with Softpath excels in length generalization at ListOps with moderate performance on argument generalization, whereas Ordered Memory excels (comparatively) in argument generalization while struggling in length generalization."
>
> "Softpath and Gumbelpath helps in ListOps and IMDB over base BT-Cell however hurts in systematicity in logical inference."
>
> "The ideal future direction would be to extend to methods which generalize better in all aspects [...]"
>
> We were explicit about the trade offs involved here in different variants of BT-Cell instead of abstracting them away and we were explicit about the open problem that there is no one uniform model that is superior (or on par) to all other models in all the evaluated factors. Thus the suggestion for "the ideal future direction".
>
> Given that we were proposing different variants of top-k operators to apply here, we believe it obligates us to show the results of those different variants. Moreover, our motivation of showing beam=2 results was mainly to explore how well one can do with a minimally costly version of beam search (we clarify this in our current version above 4.1). Furthermore, the point of "hyperparameter fine tuning" is orthogonal to discrepancies in model results. One could make the same critique if one specific hyperparameter set were superior in all aspects. However, this kind of critique would also apply to any paper that shows results from different variations of a model (eg. ablations). Either way, we can also say that RandomTreeGRC, BalancedTreeGRC, RecurrentTreeGRC, GumbelTreeGRC, GumbelTreeLSTM etc. are all variations of the same framework - just switching the scoring function. So the baselines can be unfair too by the same token. Moreover in synthetic datasets, it is important to note that even our worst models outperforms most baselines except OM/CRvNN.

---

### Official Review · Reviewer_bsU1 · 2022-10-24

**Confidence:** 5
**Correctness:** 3
**Technical Novelty And Significance:** 2
**Empirical Novelty And Significance:** 2
**Recommendation:** 5

**Clarity, Quality, Novelty And Reproducibility:**

**Clarity and Quality**: While this paper is generally clear to me, the following points could be improved:
- The steps in Section 3 could be very hard to digest for those familiar with neither Choi et al. (2018) nor Chowdhury & Caragea (2021). More details with equations would be helpful, especially the ones describing how two children nodes are combined into their potential parent. Here are some additional concrete ideas and questions for presentation:
  - How is each beam represented? If I understood correctly it should be something like a list of (node, score) pairs, where the number of nodes is determined by how many composition steps have been taken as of now. Is this correct? If so, I think it would be helpful to include precise math formulation.
  - Including the form of the score function would be good, or at least define the input and output space of the score function.


**Novelty**: Most of the techniques exist before. The contribution of this paper is to combine the idea of beam search with latent tree structure learning in recursive neural networks, and demonstrate the effectiveness on a synthetic dataset.

**Reproducibility**: The authors have included the code and experiment details in their supplementary material. While I haven't had a chance to try it myself, I believe these materials are sufficient enough to reproduce the results.


**Strength And Weaknesses:**

## Strengths

- The proposed BT-cell shows improved performance on listops long sequences, showing a potential on the length-generalization in real NLP applications.

- Very comprehensive survey and comparison to existing work.

## Weaknesses

- Lack of qualitative analysis on failure cases: while the improvement by the proposed BT-cell is marginal on listops, it would be good to understand which types of sequences BT-cells help to process.

- Most techniques exist in prior work, while lack of gain on real NLP tasks.

- Missing reference: [1] applies a CKY style algorithm for CCG induction, which improved the performance of generalization on two tasks. Their expected execution results is essentially in the same spirit as this work's list of beam, and both pieces of work focus on generalization.

[1] Mao et al., 2021. [Grammar-Based Grounded Lexicon Learning](https://proceedings.neurips.cc/paper/2021/file/4158f6d19559955bae372bb00f6204e4-Paper.pdf). In NeurIPS.


**Summary Of The Paper:**

This paper presents BT-Cell, which uses a beam-search style technique to calculate the representation of a sequence with recursive neural networks while automatically determine the best backbone structure. Experiments show improvements in some generalization splits on a synthetic dataset (listops), and results on real datasets are on par with existing state-of-the-art methods.


**Summary Of The Review:**

This paper presents BT-Cell, which uses a beam-search style technique to calculate the representation of a sequence with recursive neural networks while automatically determine the best backbone structure. The contribution of this paper is to combine the idea of beam search with latent tree structure learning in recursive neural networks, and demonstrate the effectiveness on a synthetic dataset. While the proposed BT-cell achieves improved performance on listops long sequences, showing a potential on the length-generalization in real NLP applications, most techniques exist in prior work. The paper also lacks of qualitative analysis or gain on real NLP tasks.

I hereby acknowledge the value of this work, but recommend a rejection for this paper as I don't think the content and novelty is sufficient enough for ICLR.

---

> ### Author Response · Authors · 2022-11-18
> **Response 3**
>
> Thank you for your great suggestions!
>
> > -   Lack of qualitative analysis on failure cases: while the improvement by the proposed BT-cell is marginal on listops, it would be good to understand which types of sequences BT-cells help to process.
>
> Note that exact qualitative analysis in natural language data is hard to do. We can always find some failure cases but it can be hard to always find meaningful patterns (not spurious patterns in cherry-picked anecdotes) in them. Extensive analysis isn’t also the norm in other RvNN papers [1, 2] (beyond showing a few anecdotes of parses - which we also added in our latest version - see appendix E7); and often proper analysis of those aspects require separate dedicated papers like [3]. We would argue our attempts to show OOD performance in length generalization, contrast sets, counterfactual sets is already unique and more extensive than prior latent-tree papers [1,2].
>
> On the other hand, when it comes to synthetic data, we already present quantitative analysis in different splits in terms of different factors of variations (length, operators, arguments, systematicity, depth generalization). This already provides an error analysis (shows insights about what classes of samples the model performs well and where it performs worse).
>
> Moreover, in our latest versions, we have also added results in MNLI, results in some stress test splits (see Appendix E5) and tree parses along with score distributions for further qualitative insight (see Appendix E7).
>
>
> > -   Most techniques exist in prior work, while lack of gain on real NLP tasks.
> > The contribution of this paper is to combine the idea of beam search with latent tree structure learning in recursive neural networks, and demonstrate the effectiveness on a synthetic dataset.
>
> Please see our General Response particularly the section "BT-Cell novelty” (General Response 1) and "Softpath novelty” (General Response 2). We would also like to highlight our empirical contributions (see ``Empirical Contributions” in our General Response 2).
>
> > More details with equations would be helpful, especially the ones describing how two children nodes are combined into their potential parent. Here are some additional concrete ideas and questions for presentation:
>
> Thank you for these suggestions:
>
> * In section 2, we added a more explicit mathematical formalization of the sentence-encoding problem
> * In section 2.1 we added the mathematical form of the cell function (the exact GRC function was always specified in the Appendix B)
> * In the beginning of section 2.2, we have added mathematical formalization of the score function.
>
>
> > -   Missing reference: [1] applies a CKY style algorithm for CCG induction, which improved the performance of generalization on two tasks. Their expected execution results is essentially in the same spirit as this work's list of beam, and both pieces of work focus on generalization.
>
> We have included this reference among other references to using RvNN frameworks for compositional generalization. However, we don't really claim any novelty in using softmax for merging beams (that's just attention) and there are countless works that use softmax-attention to merge representations.
>
> >  -   How is each beam represented? If I understood correctly it should be something like a list of (node, score) pairs, where the number of nodes is determined by how many composition steps have been taken as of now. Is this correct? If so, I think it would be helpful to include precise math formulation.
>
> A beam should be of a form like: "((x1,x2,x3,...,xn), s)" where s is a scalar score for the whole beam, and x1,x2 etc. each  are vector node representations. Yes, how many nodes are remaining would depend on how many composition steps are taken. In the end there would remain only one node. We have added a visualization in Figure 1. Also note that the description of the algorithm is ideally meant to be supplemented with Algorithm 2 in the Appendix.
>
> [1] Right for the Wrong Reasons: Diagnosing Syntactic Heuristics in Natural Language Inference - McRoy et al. ACL 2019
>
> [2] Ordered Memory - Shen et al. Neurips 2019
>
> [3] Modeling Hierarchical Structures with Continuous Recursive Neural Networks - Ray Chowdhury et al. ICML 2021

---

> > ### Comment · Reviewer_bsU1 · 2022-11-18
> > **Thank you for your response!**
> >
> > I appreciate the authors' efforts to conduct additional experiments and improve the manuscript! Here's a few responses to your response:
> >
> > The MNLI results are quite encouraging, and I'd encourage the authors to put them into the main content in the next version of manuscript; however,  the contribution of `softpath` seems not very clear.
> >
> > By the way, I personally think the naming of `softpath top k` is a bit misleading, as it's not very "soft"; in contrast, the name makes me to think about saving some weighted average paths in the beams, or something similar to what is described at the beginning of the softpath top-k paragraph.
> >
> > > We can always find some failure cases but it can be hard to always find meaningful patterns (not spurious patterns in cherry-picked anecdotes) in them.
> >
> > Fair enough, I totally understand that it's hard to find meaningful patterns. Here is an idea that just came into my mind: one can probably run multiple experiments with the same hyperparameters and different random seeds, and see if the models consistently fail on some examples. If yes, there may be something hidden behind these examples.
> > Please keep in mind that this is just an idea suggestion, and that I am not requiring any additional experiments in the remaining rebuttal day -- I'm sorry if this puts the authors under any pressure.
> >
> >
> > Unfortunately, I don't think I'll be able to raise my rating at this time, primarily because I continue to believe that the contributions of beam tree and softpath, as well as their results, are insufficient for acceptance, but I'd like to thank the authors again for all their efforts.

---

> > > ### Author Response · Authors · 2022-11-18
> > > **Response 3.1**
> > >
> > > > By the way, I personally think the naming of softpath top k is a bit misleading, as it's not very "soft"; in contrast, the name makes me to think about saving some weighted average paths in the beams, or something similar to what is described at the beginning of the softpath top-k paragraph.
> > >
> > > It does keep "soft" paths. But it mitigates oversoftening by discretely selecting the top k-1 paths and softening (weighted averaging) only the last path in every instance. But perhaps, yes, a better name could be thought of because it's not as soft as some differentiable sorting strategies. We will think more about the naming.
> > >
> > >
> > > > Fair enough, I totally understand that it's hard to find meaningful patterns. Here is an idea that just came into my mind: one can probably run multiple experiments with the same hyperparameters and different random seeds, and see if the models consistently fail on some examples. If yes, there may be something hidden behind these examples. Please keep in mind that this is just an idea suggestion, and that I am not requiring any additional experiments in the remaining rebuttal day -- I'm sorry if this puts the authors under any pressure.
> > >
> > > Thank you for the suggestions. We will consider it.

---

### Official Review · Reviewer_muQo · 2022-10-25

**Confidence:** 3
**Correctness:** 4
**Technical Novelty And Significance:** 2
**Empirical Novelty And Significance:** 2
**Recommendation:** 6

**Clarity, Quality, Novelty And Reproducibility:**

- Novelty: The proposed model is a rather small increment on that of Choi et al. (2018), who present an easy-first parsing model which induces tree structure representations for its input, but without beam search decoding. The further model variants (e.g. the Softpath variant, which represents a large distribution over tree representations via a score-weighted combination of vectors) are also not especially surprising.
- Quality
	- Interpretation: The paper attempts an impressive synthesis of the space of models and evaluates them on a level playing field. However, too little space is given to analyzing and understanding the results: the conclusion and much of the result paragraphs read as simple lists of inequalities (X does better than Y on Z), with no interpretation or merely an untested gesture at an interpretation (e.g. "the memory-augmented RNN style setup in it may be more amenable for argument generalization").
	- Results: The evaluation is limited to synthetic logical reasoning tasks except for sentiment analysis (SST). It demonstrates modest (if any) improvement over existing models in the tasks (and lags behind others especially in the only naturalistic task tested). No error analysis is given to help us understand why these performance differences should be interesting.
		- Many of the cited papers in this literature evaluate on other naturalistic tasks, e.g. NLI, language modeling; or interpret the latent tree structures induced by their models / use them as unsupervised parsers. I would suggest the authors consider expanding their work to these kinds of evaluations in order for the results to be more comparable.
		- Even for the most successful results (99% accuracy on ListOps), there is no clear explanation or test showing why this model's success is worth considering, apart from a restatement of the model's design ("are able to get near perfect performance in length generalization ... because Softpath can allow gradient signals to (softly) truncated paths or beams (which would otherwise be completely truncated)").

**Strength And Weaknesses:**

- Impressive synthesis of existing recursive neural network models and evaluation on a level playing field.
- Novel model with a reasonable selection of variants, with some improvements in performance over baseline models.
- Limited interpretation and evaluation of why this model works (or why other models work).

**Summary Of The Paper:**

The authors present a novel algorithm for recursive neural network processing of sequence inputs. The algorithm combines easy-first parsing techniques with beam search in order to efficiently explore the space of possible latent tree structures during training and inference. They also present soft relaxations of the framework which lead to improved performance. They include an extensive evaluation of their model variants and many baseline models on synthetic and naturalistic language tasks.

**Summary Of The Review:**

This paper presents what seems to me a small increment on existing recursive neural network models, which yields modest performance increases. Interpretation of the models' successes are extremely limited. While I appreciate the attempt at a broad evaluation of many recursive neural network models, the results are not well synthesized beyond tables of numbers and lists of qualitative results.

On these grounds I recommend rejection. I recommend the authors work on evaluating and interpreting the model results in order to better support future model development, or push for meaningful performance improvements, especially on naturalistic language datasets.

---

> ### Author Response · Authors · 2022-11-18
> **Response 2.1**
>
> Thank you for your great suggestions!
>
> > -   Limited interpretation and evaluation of why this model works (or why other models work).
>
> We have added more interpretations/result discussion in S4.1 and also some extra discussions about other result sections.
>
> However, note that it is difficult to always say exactly why a model performs better. It's a fundamentally difficult question to answer. The criticism for lack of interpretation can be made broadly against a lot of papers. For example, it's not exactly clear why some neural networks perform worse in MCD splits [1] or why eliminating early stopping benefits compositional generalization [2] and so on. By similar token, some aspects of our results are unclear: why OM performs better in argument generalization? Why is softpath worse in systematicity? Finding a method to answer such questions robustly and generally would be breakthrough by itself and in need of a separate paper.
>
> That said, we made an attempt to do better with providing more interpretation/discussion where we can in the revised version.
>
> > The proposed model is a rather small increment on that of Choi et al. (2018), who present an easy-first parsing model which induces tree structure representations for its input, but without beam search decoding.
>
> Please check the “BT-Cell novelty” section in our General Response 1 (https://openreview.net/forum?id=sKDtBKYOdIP&noteId=9HImokro3wG).
>
> > The further model variants (e.g. the Softpath variant, which represents a large distribution over tree representations via a score-weighted combination of vectors) are also not especially surprising.
>
> The idea of soft-pooling different representations is nothing new. That is essentially an attention function. However, that's not where the contribution of softpath lies. The contribution and novelty of softpath is how this soft-pooling is contextualized and hybridized with discrete selection for tackling top-k selection (not just soft top-1). Please check the “Softpath novelty” section in our General Response 2 for more explicit discussion on its novelty (https://openreview.net/forum?id=sKDtBKYOdIP&noteId=mp1CN8J5utR).
>
> > -   Interpretation: The paper attempts an impressive synthesis of the space of models and evaluates them on a level playing field. However, too little space is given to analyzing and understanding the results: the conclusion and much of the result paragraphs read as simple lists of inequalities (X does better than Y on Z), with no interpretation or merely an untested gesture at an interpretation (e.g. "the memory-augmented RNN style setup in it may be more amenable for argument generalization").
>
> The gesture was implicitly justified by elimination of other plausible reasons. OM can't be said to be performing so well just for being a better parser (GoldTreeCell uses gold trees but still worse than OM sometimes). It's also not performing well for using GRC which is used by many other models. So, abductively, the only other plausible reason that we could think of is the inductive bias present in its particular kind of memory-augmented setup that OM uses. Although we admit it's a bit vague and that we don't know the exact reason (again as we said before finding exact reasons it's a broader issue). We have, however, elaborated more on these claims in the latest version of the paper (please see 4.1,4.2).

---

> ### Author Response · Authors · 2022-11-18
> **Response 2.2**
>
> >  Results: The evaluation is limited to synthetic logical reasoning tasks except for sentiment analysis (SST). It demonstrates modest (if any) improvement over existing models in the tasks (and lags behind others especially in the only naturalistic task tested). No error analysis is given
>
> >  -   Many of the cited papers in this literature evaluate on other naturalistic tasks, e.g. NLI, language modeling; or interpret the latent tree structures induced by their models / use them as unsupervised parsers. I would suggest the authors consider expanding their work to these kinds of evaluations in order for the results to be more comparable.
>
> * We added MNLI results with some results on stress test splits in Appendix E5. We also added parse tree analysis in Appendix E6. Moreover, besides SST, IMDB results were also present in Table 2, including OOD test results (e.g., contrast set and counterfactuals).
> * While in ListOps CRvNN is close to BT-GRC+softpath in length generalization, BT-GRC is more-than-modestly better than CRvNN in argument generalization (CRvNN gets 24.35 in 15 arguments split, BT-GRC gets 63.7). On the other hand, while OM is better in argument generalization, BT-GRC+softpath is more-than-modestly better than OM in length generalization in ListOps (OM gets 76.9 in the 900-1000 split; BT-GRC+Softpath gets 97.2). It is important to note that there is no single model which is better or only-modestly-worse than BT-GRC+Softpath in all aspects. Instead, BT-GRC provides different but significant trade offs compared to other strong contenders like CRvNN and OM.
> * The problem of mixed results in natural language tasks applies more broadly to this specific research area. Particularly, works that do focus on tackling these synthetic tasks often end up providing limited performance boost [3][4][5] on natural language tasks (or no report on natural language tasks at all [6]). This is also partly because it is difficult to do well in both domains in a scalable manner - that is, models which are reported to do better on natural language tasks often turn out to have problems in more of the algorithmic style of tasks that are more explicitly structure-sensitive and potentially less susceptible to shortcut learnings.
>
> > -   Even for the most successful results (99% accuracy on ListOps), there is no clear explanation or test showing why this model's success is worth considering, apart from a restatement of the model's design ("are able to get near perfect performance in length generalization ... because Softpath can allow gradient signals to (softly) truncated paths or beams (which would otherwise be completely truncated)").
>
> In our opinion, any model's success should be worth considering. ListOps length generalization splits can offer a sanity check for a model's length generalization capacity. Note that ListOps, despite being synthetic, is still a language processing task which we can rephrase in natural language format: "what is the solution of maximum of 5, 6 and (7+9)". Inability to do ListOps suggests inability to process analogous kinds of natural language tasks.
>
> Overall similar to [5], we are taking a bottom up approach here. To quote [5]:
> "Particularly in algorithmic tasks, it is often the case that a sub-optimal choice of architecture/optimization method makes the model fall back to simple memorization. We argue that it is crucial to look at isolated problems which test specific generalization capability. This calls for a bottom-up approach: building on toy tasks that focus on individual aspects of generalization and using them for improving models."
>
> We are following a similar motivation - thus, our focus on synthetic tasks. But we also provide some NLP tasks to show that our models are not completely broken in those tasks.
>
> The "restatement of the model's design" was a callback to the main motivation for the extension -- to go beyond the bottleneck, created by plain top k, of limiting gradient propagation only through the final k beams. The results provide empirical support for the prior motivation. We have discussed this more explicitly in the latest version (4.1)
>
>
> [1] Measuring Compositional Generalization: A Comprehensive Method on Realistic Data - Keysers et al. ICLR 2020
>
> [2] The Devil is in the Detail: Simple Tricks Improve Systematic Generalization of Transformers - Csordás et al. EMNLP 2021
>
> [3] Ordered Memory - Shen et al. Neurips 2019
>
> [4] Modeling Hierarchical Structures with Continuous Recursive Neural Networks - Ray Chowdhury et al. ICML 2021
>
> [5] Cooperative Learning of Disjoint Syntax and Semantics - Havrylov et al. NAACL 2019
>
> [6] The Neural Data Router: Adaptive Control Flow in Transformers Improves Systematic Generalization - Csordás et al. ICLR 2022

---

> ### Comment · Reviewer_muQo · 2022-11-23
> **Response to rebuttal**
>
> I appreciate the authors' extremely thorough responses and revisions to the paper -- in particular, the addition of another naturalistic language dataset, the emphasis on the novelty of the Softpath modification, and the improved interpretation work done in section 4. I am convinced that the Softpath+beam search contribution is meaningfully novel, and while the resulting numbers aren't outstanding, it is useful for the community to better understand the full space of empirically valuable model classes. I also grant that my standards for interpretation are a bit too high for this subfield/community.
>
> On this reasoning I revise from 3->6.

---

### Official Review · Reviewer_Dk8K · 2022-10-29

**Confidence:** 3
**Correctness:** 4
**Technical Novelty And Significance:** 3
**Empirical Novelty And Significance:** 3
**Recommendation:** 6

**Clarity, Quality, Novelty And Reproducibility:**

The paper is clear overall and I believe the work is original, novel, and reproducible.

**Strength And Weaknesses:**

*Strengths*

The accuracy evaluation is thorough, and the results have convinced me that the method is performant. However, one of the biggest drawbacks of latent structure methods is scalability. I would like to better understand the computational complexity of the method and baselines.

*Weaknesses*

The paper needs an analysis of computational complexity. I believe the method has $O(B^2k)$ runtime. How do the other methods compare? A table with asymptotic runtimes / space complexity is a crucial missing component. An empirical study of the runtime (time per iteration vs sentence length) would greatly improve a scalability argument as well.



*Comments*
* The text description of beam search can be placed in the appendix.
* An illustration of the easy-first beam search with Softpath would be a nice figure to have.
* The paper is missing a citation for easy-first beam search [1].
* What is the connection between state space models and beam search mentioned in the last sentence of the paper?
* I believe beam search + softpath can be interpreted as a continuous relaxation of the sum-and-sample estimator [2], generalized to beam search. The sum-and-sample estimator takes the top-k elements from a proposal distribution and samples an extra element to eliminate bias at the cost of added variance. Instead of sampling to reduce bias, Softpath makes a soft decision that reduces bias less than a sample would but does not add variance.

[1] Ji Ma, Jingbo Zhu, Tong Xiao, and Nan Yang. 2013. Easy-First POS Tagging and Dependency Parsing with Beam Search. In Proceedings of the 51st Annual Meeting of the Association for Computational Linguistics (Volume 2: Short Papers), pages 110–114, Sofia, Bulgaria. Association for Computational Linguistics.

[2] Liu, Runjing et al. “Rao-Blackwellized Stochastic Gradients for Discrete Distributions.” ICML (2019).

**Summary Of The Paper:**

This paper presents a differentiable easy-first beam search for structure induction, called Beam Tree Recursive Cells (BT-RC). A couple methods are presented to handle the sparse gradient issue in beam search. There are two sources of sparsity: parent composition and the topk filtering of beams. The two groups of methods presented are Gumbel-BT-GRC, which uses straight-through Gumbel topk for parent composition, and Softpath variants, which use a convex combination of beam elements. The models are evaluated against strong structure-sensitive baselines and perform comparably and in some cases favorably.

**Summary Of The Review:**

I advocate for a weak reject. The method approaches key issues in scaling latent structured models, but is missing crucial analyses. I will be happy to increase my score to accept given more analysis on space/time complexity and empirical speed numbers.

Edit: After updated score from 5->6 after author's response.

---

> ### Author Response · Authors · 2022-11-18
> **Response 1**
>
> Thank you for your great suggestions!
>
> > The paper needs an analysis of computational complexity. I believe the method has O(B2k) runtime. How do the other methods compare? A table with asymptotic runtimes / space complexity is a crucial missing component. An empirical study of the runtime (time per iteration vs sentence length) would greatly improve a scalability argument as well.
>
> We have added efficiency analysis in Appendix E6. We focused more on empirical efficiency analysis because asymptotic analysis can fail to reflect practical performance given that it doesn’t concretely reflect which part is parallelizable in CUDA and which part has to be necessarily sequentially processed.  But overall, we have attempted a very detailed discussion in E6 over relevant efficiency concerns with some elements of asymptotic analysis as well.
>
> > -   The text description of beam search can be placed in the appendix.
>
> We have pushed the main description to the appendix.
>
> > -   An illustration of the easy-first beam search with Softpath would be a nice figure to have.
>
> We have added Figure 1 for visualization.
>
> > -   The paper is missing a citation for easy-first beam search [1].
>
> Thank you for pointing it out. We have added the citation.
>
> > -   What is the connection between state space models and beam search mentioned in the last sentence of the paper?
>
> We are not yet positing a concrete idea. But SSMs can be interpreted as a form of linear recurrence [1] and can be utilized to speed up recurrence. Given the connection between RvNNs and RNNs (section 2), there could be some analogous way to linearize the recursion while maintaining a hierarchical composition pattern.  However, doing it in a manner so as to get the benefits of S4 [2] style computation will be a challenge. Although, initial steps to take would be to probably explore simpler forms of linearization (eg. QRNNs, SRU, GLIR) in a recursive framework.
>
> > -   I believe beam search + softpath can be interpreted as a continuous relaxation of the sum-and-sample estimator [2], generalized to beam search. The sum-and-sample estimator takes the top-k elements from a proposal distribution and samples an extra element to eliminate bias at the cost of added variance. Instead of sampling to reduce bias, Softpath makes a soft decision that reduces bias less than a sample would but does not add variance.
>
> Indeed, there are some analogies between [2] and softpath in that both of them treat higher probability items differently from the lower probability items. However, there are some substantial technical differences between the two:
>
> * [2] concentrates on computing expectations over multiple categories. A top-k generalization would be to separately calculate expectations for each of the selected $k$ items using the method in [2]. So then the top $r^{th}$ selected item (where $r \leq k$) would involve calculating the expectation based on the probability distribution for any given item being the top $r^{th}$ one. Then we can approximate the expectation using [2]. However, softpath doesn't try to create expectations for the top $k-1$ selections. It just selects them discretely based on argsort without even computing probabilities for being the $r^{th}$ item. And while it creates a sort of expectation for the $k^{th}$ selection over the remaining bottom items, it computes the full expectation using softmax (just plain attention) instead of using [2].
> * The problems we are tackling are also very different. [2] is concerned with creating expectations efficiently when there is a large number of items. However, in our case, at any particular instance of Softpath, our choices are bound to be around $k^2$ where $k$ is the beam size (which would practically range between 2 to 8; thus at most there would be around $64$ choices to deal with so the motivation for [2] does not apply here).
> * We have also distinguished our motivations better in the paper (while introducing softpath in page 5).
>
> [1] Combining Recurrent, Convolutional, and Continuous-time Models with Linear State-Space Layers - Gu et al. Neurips 2021
> [2] Liu, Runjing et al. “Rao-Blackwellized Stochastic Gradients for Discrete Distributions.” ICML (2019).
> [3] Efficiently Modeling Long Sequences with Structured State Spaces - Gu et al. ICLR 2022

---

> > ### Comment · Reviewer_Dk8K · 2022-11-20
> > **Asymptotic analysis and relationship to sum and sample**
> >
> > Thanks for the response. The empirical time and space numbers are worth including in the main text. Following [1], you could compare beam tree models directly to OM and CRvNN only in the main text, and also leave the full table in the appendix. I have increased the score to 6.
> >
> > I also recommend plotting the Pareto frontier of accuracy vs efficiency: a scatter plot with empirical time on the x-axis and performance (accuracy) on the y-axis, with one dot per model.
> >
> > For the asymptotic analysis, rather than a prose description I recommend reporting the big-O complexities in a table with the following columns: model, serial time complexity, parallel time complexity, and space complexity (either at training or test time).
> >
> > Sorry for not being clear about the relationship to sum and sample. The sum and sample estimator is very general, and the beam+softpath method could/should also be described in a general setting. Two potential ways beam+softpath improves over sum and sample: 1) Beam+softpath takes advantage of recursive structure and 2) softpath improves variance at the cost of bias. Investigating and quantifying these connections would be nice to see in future work.
> >
> > Finally, I believe this paper would go from good to great with more focused writing. The method itself is compelling and experiments are thorough. The current experimental results demonstrate that BT-GRC performs as well as other models in terms of accuracy. In order to differentiate BT-GRC, a careful analysis of empirical speed and computational complexity is likely key. To make space for this, I recommend relegating some of the baselines and model variants in tables 1 and 2 to the appendix. This would also allow the writing in the results section to be tightened up and moved partially into the appendix.
> >
> > [1] Chowdhury, Jishnu Ray and Cornelia Caragea. “Modeling Hierarchical Structures with Continuous Recursive Neural Networks.” International Conference on Machine Learning (2021).

---

> > > ### Author Response · Authors · 2022-11-26
> > > **Thank you**
> > >
> > > Thank you for the clarifications and the new suggestions. We will try to incorporate them in the next version of the paper.

---

### Author Response · Authors · 2022-11-18
**General Response Part 1**

We are very grateful to all the reviewers for their constructive feedback. We made a few changes to address all the concerns made by reviewers.

**Complexity Analysis**

We have added a detailed discussion about computational efficiency and demands in Appendix E6

**Qualitative Analysis**

We have added qualitative analysis of extracted parse structures from BT-Cell models with varying beam size and top-k operators in Appendix E7.

**Additional NLP results**

We have added experiments on MNLI and some of its stress tests [1] in Appendix E5.
We show that although BT-GRC/BT-GRC+Softpath are more robust in stress sets like length matched/mismatched (LenM, LenMM) and negation matched/mismatched (NegM, NegMM) compared to other models.

**BT-Cell novelty**

A few comments from some reviewers have been made about the lack of novelty for BT-Cell.
We agree that in technical terms, BT-Cell is incrementally novel. In the abstract we framed it more as an “overlooked” framework. However, we would still like to highlight the significance of this contribution:

1. The tools for creating BT-Cell, particularly BT-LSTM, have been available since 2015. By that time, both TreeLSTM and beam-easy-first parsing were introduced. Note that BT-LSTM does not use gumbel-softmax and is not dependent on the introduction of gumbel softmax in 2017. Yet no work so far has explored BT-LSTM. Instead researchers attempted to extend beam search in more complex frameworks (shift-reduce parsing/CYK cells) which aren’t as effective (see Appendix E6 and Related Works discussion).

2. The performance boost from just this extension is not marginal. To give an example, GumbelTreeLSTM gets $36.8$ in ListOps $900$-$1000$ length split. Keeping everything the same, BT-LSTM gets $78.8$. The performance is further improved ($97+$) with a few minor changes.
3. ListOps is still a highly challenging dataset which has required sophisticated RL training techniques or several non-trivial architectural modifications (CRvNN, OM) for length generalization. Only two RL-free models moderately succeed in ListOps so far. We think it’s worth acknowledging here that just a simple extension over an already simple existing model (Choi et al. 2018) can compete on par with the prior SOTAs.
4. Note also that all the three best performers (CRvNN, OM, BT-Cell) still have different architectural trade-offs. None is completely superior in all aspects. We believe BT-Cell can provide a new lens for developing and building future models.

Part 2: https://openreview.net/forum?id=sKDtBKYOdIP&noteId=mp1CN8J5utR

---

### Author Response · Authors · 2022-11-18
**General Response Part 2**

Part 1: https://openreview.net/forum?id=sKDtBKYOdIP&noteId=9HImokro3wG

**Softpath novelty**

Our contribution is not limited to just extendeding  Choi et al. 2018 with beam search. We also introduce Softpath. We have added better motivation for the approach on Page 5. But we further highlight the significance below:

Note first that softpath is attempting to solve the problem of selecting $k$ out of  $m$ items where $m \geq k$. This immediately sets it apart from other approaches of soft-pooling, e.g., in CYK algorithms as in (Maillard et al. 2019) which simply uses attention to create a single expected representation out of many while filling chart cells. The challenges of selecting top $k$ items are different from just softly selecting top $1$ (which can be done simply by attention). For selecting top-k,  one can simply use plain top-$k$ but it sacrifices all gradient propagation through unselected paths. One can use a sophisticated differentiable sorting algorithm but it leads to washed out representations and significant slowdown (In Appendix E1, we also demonstrate much worse performance with a prior differentiable sorting algorithm and in Appendix E6 we demonstrate the slowdown from adding differentiable sorting).  Naive ideas like just adding straight-through estimation to Gumbel Top-K has its own severe problems (as we discuss in Appendix D) and also comparatively poor empirical performance (that we see in ListOps and Logical Inference).

Out of all these challenges, we are not aware of any prior work anticipating softpath as a simple hybrid approach to discretely select top $k$-1 using plain top $k$ (to keep the most promising paths mostly sharp) and softly-interpolate the bottom paths for the top kth selection. With softpath we are taking a middle way stance between creating a full soft permutation matrix leading to washed out representations  and exclusively using plain top-k altogether forgoing backpropagation through truncated paths.

Although the extension is simple (and perhaps, unsurprising in hindsight), we believe its simplicity is a virtue. Moreover, adding beam search to RvNNs have also drawn interests previously (see Related Works discussion) but none has explored beyond plain top-k in the context of RvNNs.

As such, we would contend that the idea of softpath taken as a whole (not just focusing on soft pooling operation which is just attention-pooling and nothing novel) is neither particularly trivial, nor particularly obvious (given the lack of anticipation of any analogous strategy from years of prior works despite exploring similar areas, e.g., extending models by beam search or even differentiable sorting in different contexts).


**Empirical Contribution**

We would also like to emphasize our empirical contributions. We show that ListOps is still not a completely solved task. SOTA models still suffer from argument generalization. As far as we know, this is a new discovery. We also test the robustness of RvNNs to Out-of-Distribution test sets in NLP tasks (stress tests [1] in MNLI, contrast sets/counterfactual tests, length generalization). We show that the models (CRvNN, OM, BT-Cell) that perform well in synthetic data also tend to be relatively more robust in some of the OOD tests and stress tests. As far as we know, such analysis also provides new insights for recent RvNN-based models. We also analyze and contrast the limits of recent Transformer-based models like NDR [2] in Appendix E3.

[1] Stress Test Evaluation for Natural Language Inference - Naik et al. COLING 2018
[2] The Neural Data Router: Adaptive Control Flow in Transformers Improves Systematic Generalization - Csordás et al. ICLR 2022

---

### Decision · Program_Chairs · 2023-01-20

**Decision:**

Reject

**Justification For Why Not Higher Score:**

Limited novelty, mixed results, and flaws in the experiments.

**Justification For Why Not Lower Score:**

N/A

**Metareview: Summary, Strengths And Weaknesses:**

The paper presents an algorithm (BT-Cell) for recursive neural network processing of sequence inputs, which combines easy-first parsing with beam search in order to efficiently explore the space of possible latent tree structures during training and inference. The method is an extension of Choi et al. (2018), which didn't include beam search decoding. The authors also present several variants (e.g., soft-path). The paper provides experiments both on synthetic and real data.

The discussions centered around the following three concerns:
1. After discussion, reviewers better understood the paper's differentiation from Choi et al., and the authors response on "BT-Cell novelty" and "Softpath novelty" was quite helpful. While the novelty of the paper is meaningful, the reviewers also felt that the approach is not particularly surprising.
2. The bigger concern was about the experiments, as levels of performance are relatively close to prior work and future versions of the paper should probably better highlight the accuracy vs. efficiency tradeoff. This tradeoff is currently difficult to analyze as accuracies and running times are in different tables, and the authors may want to add Pareto fronts as suggested by one of the reviewers. As the models of the paper do not particularly stand out either in terms of accuracy or speed, the reviewers thought that the empirical contribution of the paper is quite limited.
3. Reviewers were all concerned by the points raised by reviewer dDzD on the ablation of beam size k parameter on the test set. First, as the authors conceded during the discussion, the analysis of the k parameters should have been done on dev sets instead of test sets. While the authors might be right that there are “countless examples of published papers […] where ablations are done in test sets”, these published papers often put their ablations in separate tables. Instead, the authors lump ablations and comparison to prior work in the same *test-set* tables, which is problematic. Indeed, the models of the paper are sometimes shown to “win” (i.e., bold font) with k=5 on some metrics, while winning with k=2 on some other metrics. This is akin to cherry-picking k based on the column (metric) one is looking at, which is not acceptable when done on the test set and especially not in comparisons to models of prior work. Given that the authors only tried two different values of k and results between k=2 and k=5 seem correlated, the problem may not be too serious, but reviewers remained quite concerned as experimental gains are already not a strength of the paper and some of the performance increases over prior work are small and sometimes nonexistent.

Given these three reasons, the reviewers converged on a rejection recommendation during the reviewer-AC meeting, and I side with them.

**Summary Of Ac-Reviewer Meeting:**

The discussion centered on the 3 points mentioned in the meta review. All reviewers finally agreed:
1. The methods of the papers are often borrowed from prior work and otherwise not surprising. The main novelty of the paper (soft-path) is more of a "hack" according to reviewers.
2. Results are lukewarm and often hard to interpret.
3. Evaluating different values of k on test sets is unfair (at least in the way the results are presented, as 2 different values of k gives them twice as many chances to "win" against prior work).

All reviewers finally agreed to reject.